# Distinct sensorimotor feedback loops for dynamic and static control of primate precision grip

Tomomichi Oya [1,2,4], Tomohiko Takei [1,2,3,4] & Kazuhiko Seki [1,2✉]

Volitional limb motor control involves dynamic and static muscle actions. It remains elusive how such distinct actions are controlled through separated or shared neural circuits. Here we explored the potential separation for dynamic and static controls in primate hand actions, by investigating the neuronal coherence between local field potentials (LFPs) of the spinal cord and the forelimb electromyographic activity (EMGs), and LFPs of the motor cortex and the EMGs during the performance of a precision grip in macaque monkeys. We observed the emergence of beta-range coherence with EMGs at spinal cord and motor cortex in the separated phases; spinal coherence during the grip phase and cortical coherence during the hold phase. Further, both of the coherences were influenced by bidirectional interactions with reasonable latencies as beta oscillatory cycles. These results indicate that dedicated feedback circuits comprising spinal and cortical structures underlie dynamic and static controls of dexterous hand actions.

[1] Department of Neurophysiology, National Institute of Neuroscience, National Center of Neurology and Psychiatry, Tokyo, Japan. [2] Department of Developmental Physiology, National Institute for Physiological Science, Aichi, Japan. [3] Present address: Department of Physiology and Neurobiology, Graduate School of Medicine/The Hakubi Center for Advanced Research, Kyoto University, Kyoto, Japan. [4] These authors contributed equally: Tomomichi Oya, Tomohiko Takei. ✉email: seki@ncnp.go.jp

Motor behaviors comprise a continuum of "moving" a limb and "holding" it still via dynamic and static control of muscle activity. A long-standing fundamental question is whether the dynamic and static actions are achieved through similar or specialized control processes. Use of a similar control process for both moving and holding has the advantage of generality and conserved neural circuitry; use of specialized control processes has the advantages of context-dependency and flexibility. However, specialized processes require neural circuits for greater dedication and expertise[1]. Elucidation of the underlying mechanisms of control processes is the key to understanding a motor system that is confronted with conflicting demands between generality and flexibility. Specialized processes and underlying neural circuits have been demonstrated in the primate brainstem for the dynamic and static control of saccadic eye movement[2,3].

It remains unknown whether dedicated circuits for skeletomotor control are at work. So far, a clear separation of discharge patterns in the motor-related areas has not been demonstrated; although some neurons in the caudal part of motor cortex show tonic discharge in the static phase[4,5], the majority of neurons in the motor cortex discharge strongly during the dynamic phase[6,7]. Downstream premotoneuronal neurons, such as corticomotoneuronal (CM) cells in the caudal part[8] and spinal premotor interneurons, typically discharge in a phasic-tonic manner[7,9]. The phasic-tonic discharges of these cortical and spinal premotoneuronal neurons can be understood as an integrated discharge that drives spinal motoneurons (MNs) to the desired extent. Interestingly, the discharges of these premotoneuronal neurons often exhibit a lack of temporal correlation with the target muscles[10,11]. This implies that other mechanisms may complement the discharges of the premotoneuronal neurons. One possible mechanism may be neuronal synchrony, because temporal synchronization also plays a critical role in neural interactions[12,13].

Neuronal synchronization has been demonstrated between the LFP of the motor cortex and EMG at the beta frequency range (15–30 Hz). However, the coherence manifests predominantly during the sustained control of muscle action, not during the dynamic phase[14–19]. If an LFP–EMG coherence manifesting exclusively during the dynamic phase could be demonstrated in the limb-related motor structure, it could provide an important step in the elucidation for specialized processes in limb control. Furthermore, it remains unresolved whether the coherent oscillations are generated from an efferent entrainment of oscillatory cortical drive transferred to the muscles[20], or from a reciprocal interaction of motor commands and sensory feedbacks between the motor cortex and the muscles[21,22].

To address these issues, we sought a potential coherent oscillation with muscles in the spinal cord as well as in the motor cortex, since spinal interneurons receive convergent inputs from the descending pathways including the corticospinal and other tracts[23], and spinal premotor interneurons are clearly discharged in relation to the dynamic and static muscle activity[9]. Also, we sought to disambiguate the two possible mechanisms for emergence of neural coherence in a more decisive way by delineating information flows and the time lag estimates between coherent signals.

We analyzed neural coherence and information flows between LFPs from the spinal cord and the motor cortex, and EMG activity of the forearm, while the macaque monkey performed a precision grip task that involved the control of both dynamic grip and static hold. We found the emergence of significant spinomuscular and corticomuscular coherence as distinct time–frequency patterns relevant to the dynamic grip and static hold phases. Furthermore, directional information analyses indicated that spinal and cortical beta-range coherence comprised a reciprocal interaction with the muscles, with corresponding time lags for beta oscillations. Furthermore, we showed that these two feedback loops (i.e., spinal local feedback vs. cortical divergent feedback loops) differ in the muscles involved. These results indicate that distinct sensorimotor feedback loops are engaged in the dynamic and static control of precision grip of primates.

## Results

**Experiment, behavior, and analyses.** We recorded spinal or cortical LFP signals from four macaque monkeys using single microelectrodes in conjunction with EMG activity from the forelimb muscles (Fig. 1a and Supplementary Fig. 1; see Table 1 for muscle lists for each monkey) while each monkey was performing a precision grip task involving dynamic grip (grip) and static hold (hold) periods. The monkeys were instructed to acquire visual targets that represented lever positions, by pinching a pair of spring-loaded levers with the thumb and index finger (Fig. 1b). Spinal LFP signals were recorded from four monkeys. The analyzed signals included 4 LFPs and 2 EMGs from monkey U, 7 LFPs and 19 EMGs from monkey A, 72 LFPs and 20 EMGs from monkey E, and 1 LFP and 21 EMGs from monkey S. Cortical LFP signals were obtained from two monkeys: 71 LFPs and 20 EMGs from monkey E, and 26 LFPs and 21 EMGs from monkey S. LFP–EMG pairs with electrical cross-talk were excluded from the analysis (see Methods). The analyzed LFP–EMG pairs are summarized in Table 2.

**Distinct types of time–frequency coherence patterns: spinal broad-band, spinal beta-band, and cortical beta-band coherence.** To examine the overall time–frequency patterns in coherence, we applied wavelet transformation on LFP and EMG signals with respect to grip onset or release onset (from 1 s before and 0.5 s after the onset), both of which were then processed for spectral analysis (Fig. 1c–e). We found three major types of coherence patterns in the spinomuscular or corticomuscular coherence. Spinomuscular coherence exhibited two types: one was termed spinal broad-band (BB) coherence, which exhibited paralleled temporal evolution with that of the paired EMG pattern, namely phasic-tonic activity during grip–hold phases (Fig. 1c); the other was termed spinal narrow-band (NB) coherence in the beta-range, which markedly emerged during the grip phase (Fig. 1d). In contrast, corticomuscular coherence appeared as an NB in the beta-range, being pronounced specifically during the hold phase (Fig. 1e).

To classify these time–frequency patterns, we measured the following two features of coherence: (1) the integral of the contours in the wavelet coherence and (2) the frequency width of coherence in a fixed time period as a supplement that was used in a previous study[24] (Supplementary Fig. 2a; see Methods for details). The integral of the contours in the wavelet coherence during the grip (0–1 s from the grip onset) or hold phase (−1 to 0 from the release onset) was calculated as the sum of significant areas at each contour level. With this measurement, we found a bimodal distribution for spinomuscular coherence (a horizontal marginal histogram in Supplementary Fig. 2b), which was successfully separated using a Gaussian mixture model (GMM). The two separated distributions were assigned to the NB type (Supplementary Fig. 2b, blue), and the BB type (Supplementary Fig. 2b, red), respectively. As the integrated contour takes a temporal profile into consideration as well as the significance bandwidth, misclassification of putative BB coherence with limited bandwidth is avoided (lower-right quadrant in Supplementary Fig. 2b). For corticomuscular coherence we saw only unimodal distributions in both frequency width and contour

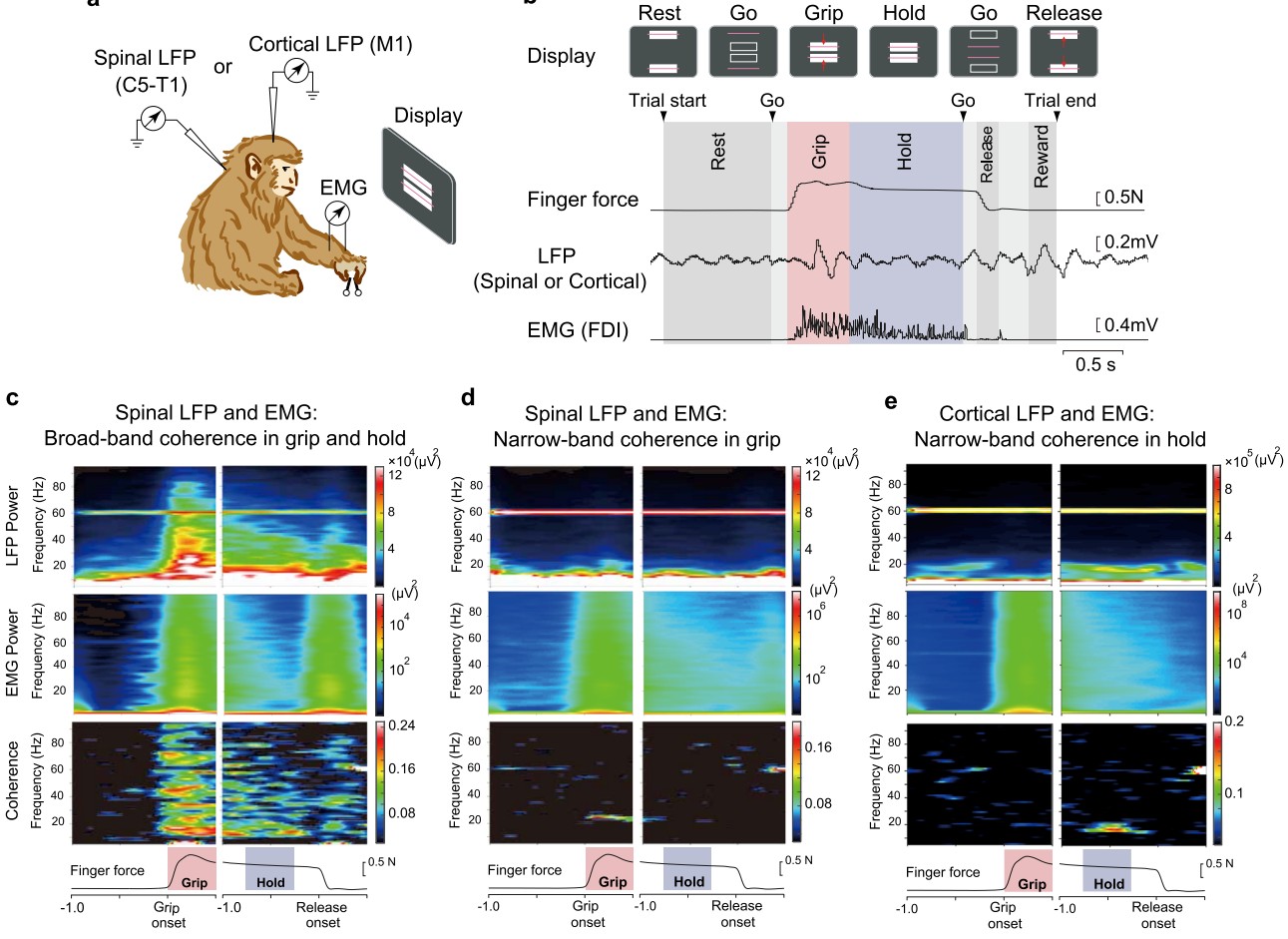

**Fig. 1 Experimental setup and representative data. a** Task and recording setup. Monkeys performed a precision grip task by squeezing a pair of spring-loaded pivoted levers to align the bars (corresponding to lever displacements) to prescribed rectangle areas in the front display. During the task either spinal or cortical local field potentials (LFPs) were recorded in conjunction with electromyography (EMG) of forelimb muscles. **b** Task epochs delineated by events, and exemplar raw traces of finger force (sum of two lever forces), LFP, and EMG. On visual and auditory cues, the monkey squeezes the levers (grip phase: red-shaded area) and maintains the force for 1–2 s (hold phase: blue-shaded area). **c–e** Representative patterns of power spectra of neural LFPs, EMGs and their coherence between spinal broad-band (BB) LFP and Abductor Pollicis Brevis (AbPB) (**c**), spinal narrow-band (NB) LFP and Abductor Pollicis Longus (AbPL) (**d**), and cortical NB LFP and Abductor Digiti Minimi (AbDM) (**e**).

integral, which led us to define all of them as NB (Supplementary Fig. 2c).

Classified spinomuscular BB coherence (97 pairs in total; 5 from monkey A, 90 from monkey E, and 2 from monkey S) manifested not only in the grip phase but also in the hold phase (Fig. 2a), with ca. 90% of pairs (5 out of 5 pairs for monkey A, 79 out of 90 for monkey E, and 1 out of 2 for monkey S, Supplementary Fig. 3a) and ca. 40% of pairs (2 out of 5 for monkey A, 37 out of 90 for monkey E, and 1 out of 2 for monkey S, Supplementary Fig. 3a) showing significant coherence during the grip or hold phases, respectively (Fig. 2d). Spinal NB coherence (88 pairs in total; 5 from monkey U, 7 from monkey A, 73 from monkey E, and 3 from monkey S, Supplementary Fig. 3b) emerged in the beta-band, predominantly during the grip phase (Fig. 2b) with more than 90% pairs (5 out of 5 pairs for monkey U, 7 out of 7 pairs for monkey A, 69 out of 73 for monkey E, and 3 out of 3 for monkey S, Supplementary Fig. 3b) showing significance exclusively during the grip phase (Fig. 2e). This contrasted with cortical NB coherence (127 pairs in total; 104 from monkey E and 23 from monkey S6), which was evident almost exclusively during the hold phase (Fig. 2c), with more than 80% pairs (92 out of 104 for monkey E and 14 out of 23 for

monkey S, Supplementary Fig. 3c), indicating significance only during the hold phase (Fig. 2f).

To further explore the functional differences among the coherence types, we examined which individual muscles were coherent with the spinal or cortical LFPs (Fig. 3). Overall, the coherent LFP–muscle pairs were observed predominantly in intrinsic hand, extrinsic hand, and wrist flexor muscles. We found fewer pairs in forearm extensors, or radial-innervated muscles (AbPL, ED23, ED45, EDC, ECR, ECU and BRD) in any coherence types. Spinal BB patterns (Fig. 3a; red) were most widely distributed among the observed muscles, showing reflection of all recruited muscles. Spinal NB coherence (Fig. 3a, b; blue) showed a preference in the index finger muscles (i.e., FDI and FDPr). This specific preference was contrasted with a relatively wider distribution of the cortical NB coherence observed in the finger muscles (Fig. 3b; yellow).

While a substantial proportion of significant pairs of both types of spinal coherence patterns were found in the grip phase, a significant difference in latency was observed with respect to grip onset (Supplementary Fig. 4). Spinal BB coherence was distributed with a median value of 95 ms prior to grip onset, whereas spinal beta-band was distributed with a median value of 7 ms prior to

**Table 1 Recorded muscles for each animal.**

| Muscle | Monkey U | Monkey A | Monkey E | Monkey S |
|---|---|---|---|---|
| Adductor Pollicis (ADP) | | x | x | x |
| First Dorsal Interosseous (FDI) | x | x | x | x |
| Second Dorsal Interosseous (2DI) | | | x | x |
| Third Dorsal Interosseous (3DI) | | | x | x |
| Fourth Dorsal Interosseous (4DI) | | | x | x |
| Abductor Pollicis Brevis (AbPB) | | x | x | x |
| Abductor Digiti Minimi (AbDM) | x | x | x | x |
| Flexor Digitorum Superficialis (FDS) | | x | x | x |
| Flexor Digitorum Profundus, radial part (FDPr) | | x | x | x |
| Flexor Digitorum Profundus, ulnar part (FDPu) | | x | x | x |
| Palmaris Longus (PL) | | x | x | x |
| Flexor Carpi Radialis (FCR) | | x | x | x |
| Flexor Carpi Ulnaris (FCU) | | x | x | x |
| Abductor Pollicis Longus (AbPL) | | | x | x |
| Extensor Carpi Radialis (ECR) | | x | x | x |
| Extensor Carpi Ulnaris (ECU) | | x | x | x |
| Extensor Digitorum Communis (EDC) | | x | x | x |
| Extensor Digitorum-2,3 (ED23) | | x | x | x |
| Extensor Digitorum-4,5 (ED45) | | x | | |
| Brachioradialis (BRD) | | x | x | x |
| Biceps Brachii (Biceps) | | x | x | x |
| Triceps Brachii (Triceps) | | | x | |

**Table 2 Recorded and analyzed LFP and EMG pairs in studied structures for each animal.**

| Structure | Spinal cord | | | | Motor cortex | |
|---|---|---|---|---|---|---|
| Monkey | U | A | E | S | E | S |
| LFP recordings | 4 | 7 | 72 | 1 | 71 | 26 |
| EMG recordings | 2 | 19 | 20 | 21 | 20 | 21 |
| LFP–EMG pairs | 8 | 133 | 1440 | 21 | 1420 | 546 |
| EMG cross-talk | 0 | 73 | 240 | 12 | 320 | 111 |
| Analyzed pairs | 8 | 60 | 1200 | 9 | 1100 | 435 |

grip onset ($t$ test with unequal variance, $p = 1.2 \times 10^{-7}$). Spinal BB coherence occurs first, leading to spinal beta-band coherence within ca. 90 ms.

**Differences in frequency, phase, and intermuscle connections between spinal and cortical coherence in the beta-range.** We then investigated whether a difference exists between the spinal and cortical NB patterns, by comparing frequency and phase distributions between the two coherence patterns (Fig. 4). The spinal NB coherence showed a slightly higher frequency content in the normalized frequency distribution than the cortical NB coherence (Fig. 4a, b). We also found each phase distribution clustered to a specific angle (red lines in Fig. 4c, d, Rayleigh test, $p = 2.2 \times 10^{-9}$ for spinal NB, $p = 5.1 \times 10^{-18}$ for cortical NB). The spinal NB lagged behind the muscle activity (Fig. 4c), whereas the cortical NB occurred prior to the muscle activity (Fig. 4d). These two distributions were statistically different (Mardia–Watson–Wheeler test, $p = 8.8 \times 10^{-10}$). These results indicate that the spinal and cortical NB coherence patterns may have arisen from different interaction processes, albeit in close frequency bands.

We frequently observed coherence between an LFP at a given recording site and multiple EMGs, which may imply an interaction among the muscles through a shared network. We defined the muscles that simultaneously emerged at a given site as "interacting muscles" mediated by spinal or cortical NB coherence, and counted the number of combinations of those muscles for each spinomuscular and corticomuscular coherence. A marked contrast was found in the intermuscle connections between the spinal and cortical NB coherence; for spinomuscular coherence (104 pairs in 21 sites), the interacting muscles were predominantly clustered in the forearm flexors (extrinsic hand and wrist flexors) (Fig. 5a). In contrast, for corticomuscular coherence (59 combinations in 28 sites), there were divergent connections among the muscles, ranging from the intrinsic hand muscles to upper arm muscles (Fig. 5b).

**Direction of causality in spinomuscular and corticomuscular coherence.** We further examined whether the observed coherence reflects putatively causal interactions in a particular direction. To explore the direction of an influence and its phase–lag relationship between the neural structures and muscles, we used a combination of directed and partial directed coherence measures based on Granger causality and a multivariate autoregressive (MVAR) model. Each measure is complementary to each other in determining a causality and estimating the lag (see Methods for details). We found that spinal BB coherence predominantly comprised an efferent pathway with relatively weak beta afferent components (Fig. 6a, d), whereas spinal beta-band coherence in the grip phase comprised bidirectional interactions in the beta-range with dominant afferent components (Fig. 6b). Corticomuscular coherence comprised a beta-range bidirectional interaction between the afferent and efferent pathways with dominant efferent components (Fig. 6f). The phase delay of spinal BB coherence was ca. 8.0 (±5.6, quantile) ms for the efferent components (Fig. 7a, d), consistent with the conduction delay between the spinal MNs and the muscles[25].

For spinal NB coherence, the beta-band afferent and efferent delays were 26.8 (±14.8) and 30.2 (±7.3) ms (Fig. 7b), respectively. For cortical beta-band coherence, the delays were 27.0 (±8.7) and 25.7 (±14.1) ms (Fig. 7f). The aggregate median for the entire delay for spinomuscular and corticomuscular coherence (i.e. sums of afferent and efferent delays) was ca. to 50–60 ms, a reciprocal of the central frequency of the coherence (ca. 15–20 Hz).

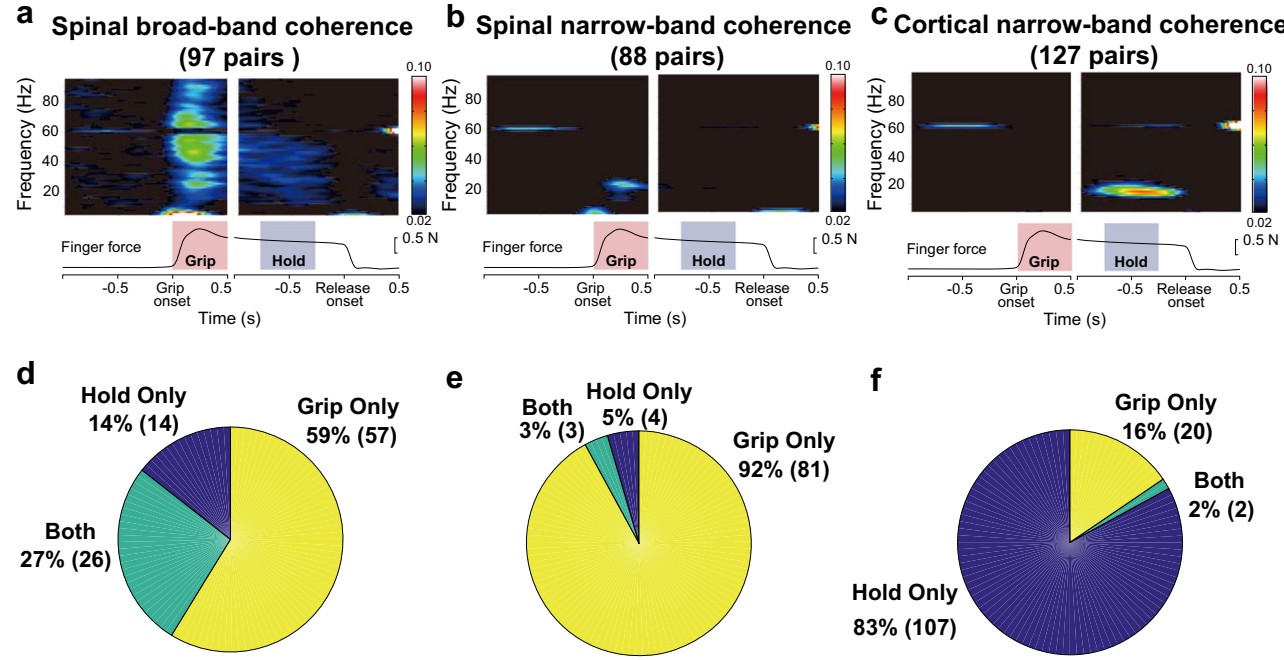

**Fig. 2 Group averages and proportions of significant pairs in grip and hold phases in each coherence pattern. a–c** Averaged wavelet coherence patterns: spinal BB, spinal NB, and cortical NB classified according to the distributions shown in Supplementary Fig. 1B. **d–f** The proportions of significant pairs in grip, hold, and both phases for the classified coherence patterns; **d** for spinal BB, **e** for spinal NB, and **f** for cortical NB.

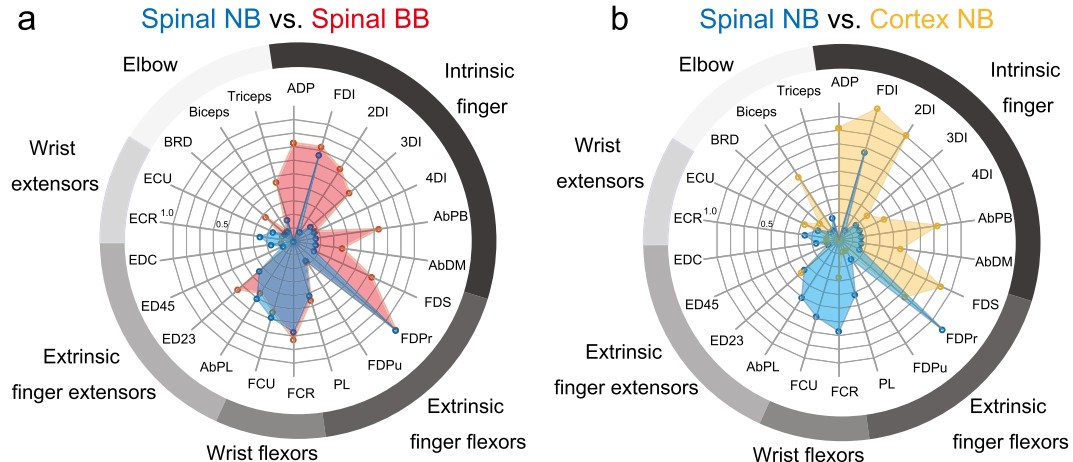

**Fig. 3 Muscle distributions of LFP–EMG coherent pairs for spinal NB (blue), spinal BB (red) and cortical NB (yellow) groups.** Muscles are ordered clockwise; intrinsic finger muscles, extrinsic finger flexors, wrist flexors, extrinsic finger extensors, wrist extensors, and elbow muscles. **a** Comparison of the distributions between spinal NB (blue) and spinal BB (red) coherent pairs. **b** Comparison of the distributions between spinal NB (blue) and cortical NB (yellow) coherent pairs.

## Discussion

It has been unclear whether distinct circuits are engaged in dynamic vs. static control of limb muscle actions. We found that the beta-range neural coherence with muscles emerged in the spinal cord and motor cortex, each of which was distinctively evident during dynamic grip or static hold phases of precision grip. Furthermore, neural information flows were bidirectional in both of beta-range spinomuscular and corticomuscular coherence with reasonable latencies for beta oscillatory cycles, indicating that dedicated feedback loops underlie each coherent pattern. The muscle groups involved in each coherence are also distinct; the trans-spinal loop involves recruitment and interactions between the index finger muscles (FDI and FDPr) and neighboring

forearm flexors, whereas the trans-cortical feedback loop arises more broadly through all the recruited finger muscles, with divergent interactions across the forelimb joints.

In our previous study, we reported BB coherence between the spinal LFP and a forelimb muscle. In light of the wide frequency–range correlation (i.e., paired LFP–EMG signals are correlated in any frequency contents), the depth of the electrode from the dorsal surface of the spinal cord, and the result of a time domain analysis on lag estimation, the coherent pattern was putatively attributed to MN pool activity[24]. However, critical evidence for this claim was lacking such as similarities in the spatiotemporal patterns and the directionality of information transfer, concomitant with more accurate lag estimation in a

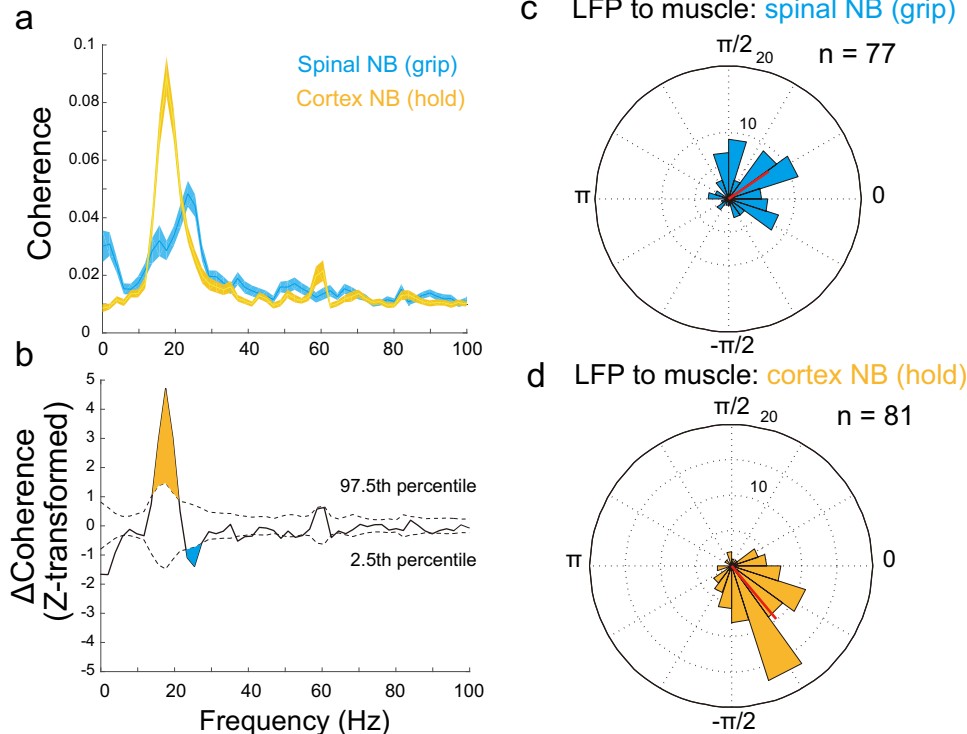

**Fig. 4 Comparison of frequency and phase distributions between the spinal NB coherence during the grip phase and the cortical NB coherence during the hold phase. a** Frequency distributions for the mean for spinal NB (blue, $n = 88$) and cortical NB (orange, $n = 127$). The peak frequency for the spinal NB is 23.4 Hz and for the cortical NB is 17.5 Hz (shaded areas are ±SEM). **b** Differences between the mean coherence values (Z-transformed). Dashed lines denote 97.5th and 2.5th confidence intervals obtained from the Monte Carlo method (10,000 iterations). The orange-shaded area represents the frequency band where cortical NB is greater than spinal NB, and blue-shaded area indicates the band where spinal NB is greater than cortical NB. **c, d** Phase distributions in significant phase–lag relationship pairs for the spinal (**c**, $n = 77$) and cortical NB (**d**, $n = 81$). Each mean angle is shown as a red vector ($0.59 \pm 1.19$ for spinal NB, $-0.87 \pm 0.91$ for cortical NB, mean ± SD) and both are nonuniformly distributed (Rayleigh test, $p = 2.2 \times 10^{-9}$ for spinal NB, $p = 5.1 \times 10^{-18}$ for cortical NB), which are statistically inhomogeneous (Mardia–Watson–Wheeler test, $p = 8.8 \times 10^{-10}$).

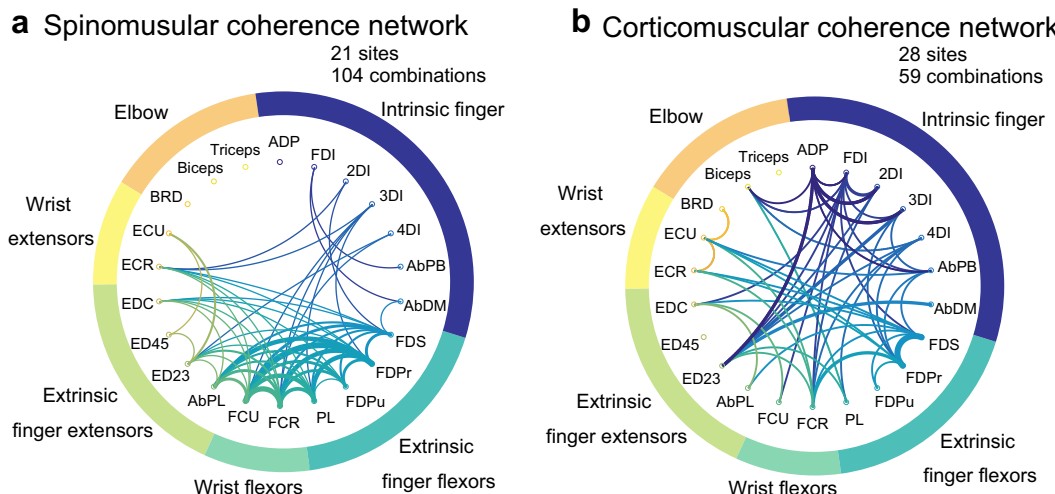

**Fig. 5 Comparison of putative intermuscle connections through spinal and cortical NB coherence.** The thickness of the line represents the number of connections observed between the muscles; the thinnest line denotes the minimum connection of a value of 1 and the thickest line indicates the maximum connection as a value of 7. **a** The connections mediated by spinomuscular NB coherence are predominantly found among synergistic muscles (extrinsic finger flexors and wrist flexors). **b** The connections mediated by the corticomuscular NB coherence are observed in diverse combinations of muscles across the forelimb, ranging from the intrinsic hand to the upper arm muscles.

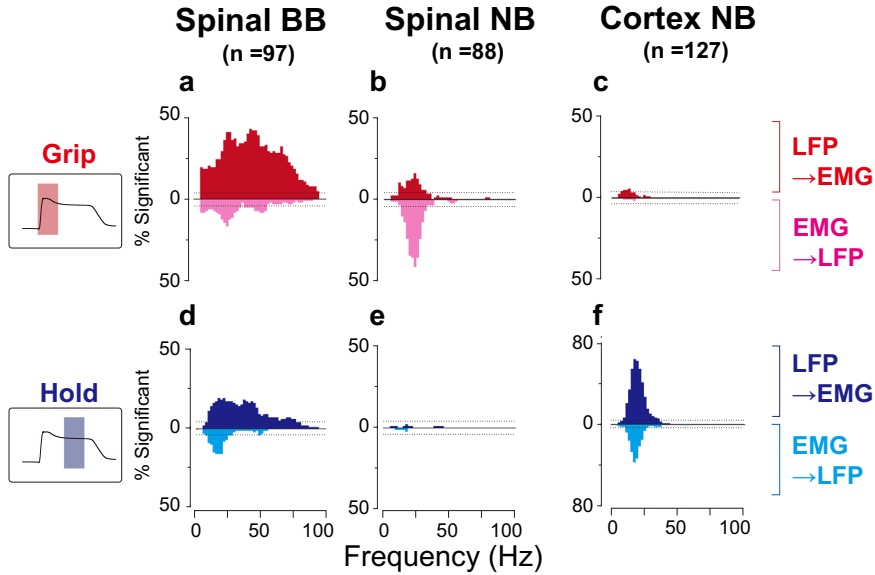

**Fig. 6 Pooled directed coherence distributions of significant frequency bands for efferent (LFP to EMG: red for grip, blue for hold) and afferent (EMG to LFP: magenta for grip, cyan for hold) components of directed coherence for grip and hold phases.** Each horizontal line represents the significance level (binomial parameter estimation with *p* = 0.005). **a–c** Spinal BB, spinal NB and cortical NB for grip phase. **d–f** Spinal BB, spinal NB and cortical NB for hold phase.

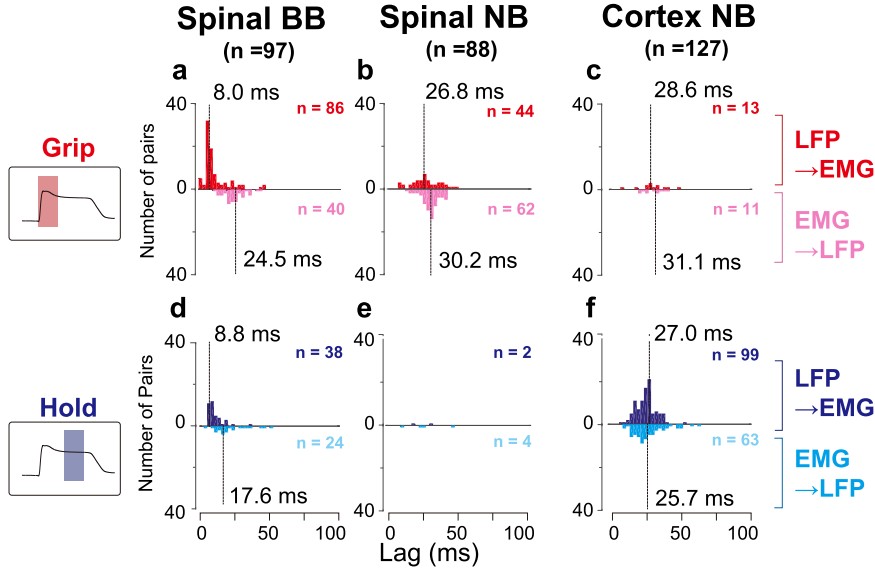

**Fig. 7 Distributions of the phase lag of partial directed coherence for efferent and afferent components.** The number of significant pairs, as determined by directed coherence analysis (Fig. 6), are shown at the upper right for each distribution. Captions and arrangements of the histograms are the same as in Fig. 6. **a–c** Spinal BB, spinal NB and cortical NB for grip phase. **d–f** Spinal BB, spinal NB and cortical NB for hold phase.

particular direction. In the present work, we found BB coherent patterns representing common temporal profiles of both LFPs and EMGs (i.e., phasic-tonic activity) (Figs. 1c, 2a, d [9]), and spatial distribution over the recruited muscles (Fig. 3a). Directional information analyses based on MVAR further indicate that the BB transfer was exclusively efferent from the spinal LFPs to the muscles (Fig. 6a, d), with a physiologically plausible lag (ca. 8–9 ms, Fig. 7a, d [25]). Consistent with our previous report, BB coherence was found at specific depths, whereas NB coherence was observed throughout the recording depths (*p* = 0.0034, χ² test, Supplementary Fig. 5 [24]). These depths are likely to

correspond to the sites where MN pools of forearm and hand muscles are located in the cervical enlargement. Collectively, spinomuscular coherence with a broad range of frequencies represents a direct transfer of a signal between the MN pool and the innervated muscles.

Most of the beta-band coherence in the spinal cord were observed during the grip phase (Fig. 2b, e). To the best of our knowledge, this is the first report on a neural structure showing noticeable beta-range coherence with the muscles during the dynamic phase of movement. We initially hypothesized that the coherent pattern would reflect convergence of efferent motor

desynchronized discharge upstream in the descending pathway[26,27] and would occur prior to the onset of muscle activity. However, the spinal beta coherence pattern lagged behind the MN pool activity, which was reflected by the BB coherence (Supplementary Fig. 4). The coherence did not consist of a unidirectional information transfer, but a bidirectional interaction with dominant afferent components to LFPs from EMGs (Fig. 6b). These findings indicate that the spinomuscular coherence evident in the dynamic phase reflects a feedback-mediated interaction between the spinal structure and the muscles, rather than a convergent efferent command relayed from the motor cortex to the muscles. The feedback is likely induced by mechanical events associated with the contraction of a muscle (e.g., a stretch of skin and a length and tension change of the contracting muscle). It appears the latencies of the afferent components correspond to that of the peak responses of spinal interneurons to a mechanical perturbation against the wrist[28]. This suggests that the afferent component of the coherence reflects a somatosensory feedback, including cutaneous and proprioceptive information. Afferent components were also found in the spinal BB coherence (Fig. 6a, d), indicating that part of the somatosensory feedbacks reaches in close proximity to the MN pool. Such an interaction might reflect a monosynaptic reflex arc (e.g., Ia monosynaptic reflex). Indeed, neurons in the dorsal root ganglion defined as Ia afferent neurons exhibit a sizable magnitude of coherence through a bidirectional interaction with the innervated muscles[29]. In addition, the sum of the medians of the phase delays for the afferent and efferent components of spinal beta coherence quantitatively matched with the cycle duration in the beta-range (Fig. 7b). Taken together, beta-band spinomuscular coherence emerging through a feedback loop probably arises from cutaneous and proprioceptive receptors, which may be triggered by mechanical events associated with muscle contraction. More complete understanding would be obtained if one could find spinal premotoneuronal neurons responding to sensory stimuli that are synchronized with the beta oscillation in the dynamic phase of the grip.

Corticomuscular coherence in the beta-band was pronounced predominantly during the hold phase (Fig. 2c, f), confirming the results extensively reported in numerous previous studies[14–17,26,30]. A small but statistically significant fraction of pairs (5.5%) represents efferent components during the dynamic phase (Fig. 6c), with the averaged onset being 0.06 ± 0.14 s with respect to grip onsets. These may reflect some contribution of synchronized cortical output to dynamic control. It remains unknown how the corticomuscular coherence emerges. One hypothesis states that it reflects an efferent entrainment of oscillatory cortical drive observed at the muscle[20]. Another suggests that the coherence arises through a reciprocal interaction of motor commands and sensory feedbacks between the cortex and the muscles[21,22]. The unresolved key analysis to addressing this question is the lag estimate. The lag estimates obtained in the previous studies were widely dispersed, sometimes out of the range of the beta-band (i.e., less than 30 ms or more than 60 ms)[31,32]. This is likely due to an inherent estimation error in the directed coherence measures, because of limited ability to dissociate direct and indirect influences on the output node[33,34]. Particularly in a nested closed loop, an effect of the previous cycle, which is delayed in a fixed time, can induce phase shifts as a function of the frequency, thereby resulting in a deviation in lag estimates. The partial directed coherence measure, which was proposed as a means of distinguishing direct influences from indirect ones[35], can provide a rather accurate estimate of lag in the closed loop[34]. Using the combination of directed coherence and partial directed coherence, we obtained phase lags with tighter distributions as compared with ones reported in the

previous studies (Fig. 7f). The sum of the medians of the phase delays for corticomuscular afferent and efferent pathways quantitatively matched with the cycle duration in the beta-range. These results indicate that corticomuscular coherence is the result of a reciprocal interaction between the motor cortex and muscles in the hold phase.

The emergence of coherence seems paradoxical as it behaves in a different manner from that of neuronal discharges. The majority of the CM and spinal premotor interneurons exhibited both phasic and tonic activities and suggested that these neurons integrate dynamic and static motor commands to produce the final motor output[9,10]. However, cortical LFPs, as reflected by synchronization of neurons, are not correlated with discharge rates of individual neurons[36]. Neuronal discharge and synchronization are thought to act cooperatively to achieve an efficient neuronal transmission. In the motor cortex, it is proposed that enhanced efficacy through synchronized oscillation may contribute to reduced discharge rates[26]. In the spinal cord, an active decorrelation mechanism within interneuronal (IN) networks is reported; while spinal INs increase the firing rates, the synchronization is reduced during the active hold[37]. This decorrelation mechanism might act either to prevent excessive MN synchrony or to maintain the information capacity of the IN network. The absence of the spinomuscular coherence during the hold period may be related to this decorrelation mechanism. In the process of integration of final motor output, neuronal discharge may be enhanced via feedback-induced synchronization during the dynamic phase. In the static phase, while the spinal neuronal discharges are actively decorrelated to secure control capabilities of MN pools, cortical descending drive utilizes the feedback-related synchronization to achieve efficient motor output. Our results indicate that both spinomuscular and corticomusclar coherence emerge through separate bidirectional sensorimotor feedback loops for each dynamic and static phase. However, it remains unknown the identity of neurons participating in the networks with more detailed connections at the cell level, including an interaction of the separate cortical and spinal loops. Further studies are warranted to explore these issues.

Considering the sensory feedback loops, which engage motor outputs through the trans-spinal and trans-cortical loops, we conceived that these loops may share routes with the short- and long-latency corrective responses to a mechanical perturbation, for the research of a feedback controller utilized for motor control[38,39]. The short latency response is the spinal-mediated, the fastest (20–50 ms) response elicited by local cutaneous and proprioceptive interactions, which leads to homonymous or synergistic muscle contractions[40]. These features are congruent to those of the spinomuscular loop; its latency is comparable to the short latency response (Fig. 7b), and it conceivably emerges via cutaneous and proprioceptive feedbacks[29]. It is predominantly observed in FDI and FDPr (Fig. 3a, b), both of which are prime movers of the forefinger that stretch and contract during a precision grip. In addition, putative intermuscle interactions observed through the spinomuscular coherence are largely confined to local or synergistic muscles of FDPr (extrinsic hand flexors and wrist flexors (Fig. 5a). The long-latency response is routed via the trans-cortical pathway with a latency of 50–100 ms[41,42]. This response is modifiable in a task-relevant manner; the response is evoked to a stretch of task-defined, broader range of muscles and adjacent mechanoreceptors via a musculoskeletal interaction, and directed flexibly to the task-related muscles, even beyond the joints, to achieve functionally oriented compensation[43–45]. Corticomuscular coherence was observed in a broad range of finger muscles, including intrinsic and extrinsic hand flexor muscles (Fig. 3b), with putative connections among the muscles being divergent, as reflected by muscle combinations

simultaneously observed through corticomuscular coherence (Fig. 5b). These hand muscles are mechanically linked through various joints and tendons, and their complex mechanical interactions may elicit cutaneous and proprioceptive feedbacks across the muscles, which are concomitantly routed back to various muscles. To stabilize the grip hold, it would be necessary to respond to normal and tangential force errors by supporting the digits from various directions by co-contracting various muscles across the joints. Indeed, it has been demonstrated that the CM system, in which many cells show sustained activities during the hold period, plays a role in joint fixation by recruiting the cells with various (e.g., synergistic, fixator, and antagonistic) target muscles[46,47]. It may be noteworthy that little coherence in radial-innervated muscles was observed either for trans-spinal or trans-cortical loops, although those muscles were active both in grip and hold phases (Supplementary Fig. 1). The present study does not provide a reasonable account of the flexor-bias. Further work is warranted for the issue. Despite the lack of direct comparative evidence, it is worth noting the common characteristics between the separate trans-spinal and trans-cortical feedback loops, and sophisticated corrective responses to mechanical perturbation with respect to mediated pathways, latencies, and involved muscles. Further studies are required to directly explore these commonalities.

The separation of engaged feedback loops in the dynamic and static phases indicates phase-specific, dedicated circuits at work in the dexterous hand control. This finding is a clearer indication of the implementation of dedicated circuits for dynamic and static control in the skeletomotor system, as compared with the moderate gradation in proportions of neuronal discharge properties between the dynamic and static phases[4,5,7,9,46]. Although specific implementation of neural circuitry is not the same as saccadic eye movements accommodated in the brainstem[2,3,48], separating the circuits for each phase may be common to both eye and limb motor circuits. There is a clear difference between the oculomotor and skeletomotor circuitry; the oculomotor circuitry rests its function on internal circuits for generating sustained activity (neural integrator), and monitoring displacement (displacement integrator)[2,3,48]. Distinctively, in the skeletomotor system, feedbacks arising from sensory afferents seem to be used more for both dynamic and static control. In the dynamic phase, the trans-spinal feedback loop may contribute to accumulating motor commands in a recursive manner such that it works as if a "neural integrator", whereas in the static phase, displacement monitoring and motor adjustment may be achieved through sensory afferent feedback loops via the supra-spinal structure. These features do not exclude putative contributions from an internally generated, feedforward command that may elude coherence analyses. These different degrees of dependence on afferent information may be explained, in part, by their physical properties and interacting environments. As compared with the oculomotor system, the skeletomotor control system (developed later phylogenetically) needs to control relatively heavier, redundant multiarticulated effectors with a larger inertia under larger gravitational influences. Under these conditions, the skeletomotor control system is more susceptible to disturbance and motor noises[49,50]. With such inherent variability of outputs arising from the effector, it would be difficult to precisely anticipate how much activity would be required to displace and sustain the limb in place. To adapt for this demand, the neural system for skeletomotor control may have shifted its dependence onto feedback control by utilizing afferent information[51].

Our findings highlight that two separate feedback controllers, as reflected by trans-spinal and trans-cortical feedback loops via phase-specific coherence patterns, may also be utilized for goal-directed, voluntary dynamic and static control of grip. Although the identity of neurons and their relevant connections remain unknown, this insight potentially provides a broader framework for understanding in voluntary dynamic and static control of our body from a feedback control perspective. The insight is particularly helpful in considering functional roles of neural coherence, as it is still debated what corticomuscular coherence specifically represents; a motor command for holding the displacement[26], active sensing such as the rodent whisker system[21,52], recalibration signals[21] or neural gain mechanisms that facilitate sensorimotor interactions[13,53]. Our results lend support to functions arising from neural–muscle–neural loops, probably related to a feedback controller, and provide a direction for designing a more desirable task framework to elucidate the functional roles of the sensorimotor loop for dynamic and static motor control.

## Methods

**Dataset**. The datasets used in the present study were obtained from four male macaque monkeys consisting of three *Macaca fuscata* (monkey U: 8.5 kg, at the age of 7, monkey A: 6.8 kg, at the age of 6, monkey S: 9.0 kg, at the age of 8) and one *Macaca mulatta* (monkey E: 5.6 kg, at the age of 5). Spinal cord datasets were obtained from monkeys U, A, and E; cortical datasets were obtained from monkeys E and S, respectively. All experimental procedures described below were approved by the Animal Research Committee at the National Institute for Physiological Sciences, and National Center of Neurology and Psychiatry, Japan.

**Behavioral task**. Each monkey was trained to squeeze a pair of spring-loaded levers with its left index finger and thumb (precision grip task; Fig. 1a, b)[9,24]. The monkey was instructed to track defined targets in a step-tracking manner by squeezing the spring-loaded levers, the positions of which were displayed on a computer screen as cursors. Each trial comprised a rest period (1.0–2.0 s), lever grip, lever hold (1.0–2.0 s), and lever release (Fig. 1b). On successful completion of the trial (1 s after the release), the monkey was rewarded with a drop of apple puree. The force required to reach the target positions was adjusted independently for the index finger and thumb of each individual monkey.

**Surgical procedures**. After the monkeys had learned the required task for a sufficient time period, we performed surgeries to implant head restraints, EMG wires, and recording chambers under isoflurane or sevoflurane anesthesia and aseptic conditions. For EMG recordings from forelimb muscles, we performed a series of surgeries to subcutaneously implant pairs of stainless steel wires (AS 631, Cooner Wire, CA, USA) acutely or chronically. Specific muscle sets (ranging from intrinsic hand to elbow muscles) for each animal are listed in Table 1. For spinal recordings, we implanted a recording chamber on the cervical vertebra (C4–C7) of monkeys U, A, E and S where a unilateral laminectomy was made on the ipsilateral side of the employed hand and arm. After completion of the spinal recordings, we performed a surgery on monkeys E and S to implant a recording chamber (a circular cylinder with a 50-mm diameter) over the skull where a craniotomy was made covering a cortical area, including the hand representation of pre- and post central gyri on the contralateral side of the employed hand and arm.

**Neurophysiological recordings**. While the monkey performed the precision grip task, we recorded the LFPs from the spinal C5–T1 segments, or from the hand area of the motor cortex (Supplementary Fig. 6) through the chamber attached either on the spinal vertebrae or on the cranium by inserting a tungsten or elgiloy alloy microelectrode (impedance: 1–2 MΩ at 1 kHz) with a hydraulic microdrive (MO-951, Narishige Scientific Instrument, Japan). The recording sites were explored with the aid of positions of vertebral segments for the spinal recordings, and the geometric information adjacent to the central sulcus and electrical microstimulation for the cortical recordings. The LFP signals were referenced to a silver ball electrode placed on a surface of dura mater of spinal cord or cerebral cortex, thereafter amplified (1000 times), band-pass filtered between 0.1 Hz and 10 kHz using a differential amplifier (Model 180, A-M Systems, WA, USA), and digitized at 20 kHz. The EMGs were amplified (3000–25,000 times) and filtered (between 5 Hz and 3 kHz) using a multichannel differential amplifier (SS-6110, Nihon Kohden, Japan) and digitized at 5 kHz. Signals from the potentiometers and strain gauges attached to levers, and from the capacitive touch sensors were digitized at 1 kHz.

**Data analysis**. All subsequent analyses were carried out offline using custom-written scripts in a MALTAB environment (Mathworks, Natick, MA, USA). Only LFP–EMG pairs that had >99 trials of the data were averaged for analysis, and LFPs from the intraspinal or intracortical sites <150 μm apart were pooled to avoid redundancies resulting from propagation of the potential.

Grip onset was defined as the time at which the rate of change in the aggregate grip force (sum of forces exerted by the index finger and thumb) exceeded $2\,N\,s^{-1}$. Release onset was, likewise, determined as the time at which the rate of change in the grip force reduced below $-2\,N\,s^{-1}$.

LFPs were band-pass filtered (fourth-order Butterworth filter between 3 and 100 Hz) and downsampled to a 250 Hz sampling rate. EMGs were high-pass filtered at 30 Hz (fourth-order Butterworth filter), rectified, and downsampled to 250 Hz by averaging every 20 bins. The averaging involves a low-pass filtering with the cut-off frequency being ca. 110 Hz (Fc = 0.443/M × Fs; Fc: cut-off frequency, M: bin number, Fs: sampling frequency[54]).

Electrical cross-talk among EMGs: To exclude spurious coherence arising from electrical cross-talk among EMGs, we quantified the degree of electrical cross-talk among EMGs recorded simultaneously using a method developed by Kilner et al.[55]. Original EMGs were downsampled to 1 kHz and differentiated three times without being rectified. The preprocessed signals were then subjected to cross-correlation analysis given as:

$$r(\tau) = \frac{\frac{1}{t_{max}} \sum_{i=0}^{t_{max}} f_1(t)f_2(t-\tau) - \overline{f}_1\overline{f}_2}{\sigma_1\sigma_2},\qquad(1)$$

where $f_1$ and $f_2$ are two differentiated EMG signals, $\overline{f}_1$ and $\overline{f}_2$ are their mean values, and $\sigma_1$ and $\sigma_2$ are their standard deviations. $r$ was calculated with 25-ms lags and a maximum modulus of $r$, $|r|_{max}$, was used as an index of the extent of cross-talk. In each experimental day, $|r|_{max}$ was calculated between EMG signals of every simultaneously recorded muscle pair for a 1-min epoch. We set the significant cross-talk threshold to 0.25 for each muscle pair, and in cases where it exceeded the threshold, we randomly excluded either one of the pair from the data pool. Furthermore, to eliminate any influence from the power line, we excluded the frequency band between ±5 Hz with regard to 50 (monkeys U, A, and E) or 60 (monkey S) Hz and concatenated the neighboring frequencies.

**Time-frequency representation: wavelet coherence.** To analyze the time series containing nonstationary power at different frequencies, we employed coherence analysis between wavelet-transformed signals. LFP and EMG signals spanning either an onset of grip or an onset of release (from 1 s before and 0.5 s after each onset) were transformed using complex gabor wavelets ($\sigma = 128$ ms) (Fig. 1c–e). We thereafter calculated the coherence between the transformed LFPs and EMGs (Fig. 1c–e), as per the equation below:

$$\text{Coh}(t,f) = \frac{|\frac{1}{N}\sum_{j=1}^{N} X_j(t,f)Y_j(t,f)|^2}{P_X(t,f)P_Y(t,f)},\qquad(2)$$

where $X$ and $Y$ are time-frequency representations and $P_X$, $P_Y$ are power spectra of LFP and EMG signals calculated using the wavelet transformation.

**Coherence in a fixed time window: standard coherence.** To compare the coherence measures in the grip and hold phases, we took time windows from 0 to 512 ms after grip onset for the grip phase and from 768 to 256 ms prior to the release onset for the hold phase for analysis. Thereafter, 128-point time series were divided into nonoverlapping segments for Fast Fourier Transform. This allowed investigation of spectral measurements with a frequency resolution of 1.95 Hz. We then calculated one-sided power spectra for the 128-point time series of LFP and EMG signals. Denoting the Fourier transform of the $i$th section of LFPs and EMGs as $F_{1,j}(f)$ and $F_{2,i}(f)$, respectively, the power spectrum of each signal ($j = 1, 2$) was calculated as:

$$P_j(f) = \frac{2}{256^2 L}\sum_{i=1}^{L} F_{j,i}(f)F_{j,i}^*(f),\qquad(3)$$

where $L$ is the number of data segments available and * denotes the complex conjugate[56]. Using this normalization, $P(f)$ has units of $\mu V^2$. The calculation of coherence between an LFP–EMG pair is as follows:

$$\text{Coh}(f) = \frac{|\sum_{i=1}^{L} F_{1,i}^*(f)F_{2,i}(f)|^2}{\sum_{i=1}^{L} F_{1,i}^*(f)F_{1,i}(f)\sum_{i=1}^{L} F_{2,i}^*(f)F_{2,i}(f)}.\qquad(4)$$

A significance threshold level $S$ was calculated according to Rosenberg et al.[57] as

$$S = 1 - \alpha^{\frac{1}{L-1}},\qquad(5)$$

where $\alpha$ is the significance level. Because we were more interested in detecting coherence bands, spurious point-wise significance had to be excluded. Thus, we put a more stringent level for the probability, i.e., $\alpha = 0.005$, corresponding to a threshold coherence value $S$ of 0.0409.

**Classification of coherence types.** To classify qualitatively different time–frequency patterns of LFP–EMG coherence (Fig. 1c–e), we quantitatively characterized those patterns based on two features of coherence measures: an integral of contours of wavelet coherence and a frequency width of standard coherence (Supplementary Fig. 2a). The integral of contour was quantified as a volume that exceeded a significant level in a time window from 0 to 1 s with regard to the grip onset for "grip", or a time window from −1 to 0 s from the onset of

release for "hold", thereby reflecting coherence strength, how wide its significant frequency band distributes, and how long the coherency extends over time. To dissociate two types of spinomuscular coherence, we then applied the Expectation–Maximization (E–M) algorithm to the distribution of integral of contours, under a GMM assumption[58]. Points assigned as NB in the contour integral dimension were examined for outliers in the frequency width dimension (Smirnov–Grubbs test, $p < 0.05$). Three points were determined as outliers and exceeded 25 Hz in frequency width, a criterion used for classification in the previous study[24]. We therefore assigned those points to the BB category.

**Comparison of latencies from the grip onset between spinal BB and NB coherence.** To examine the difference in latency with regard to the grip onset between spinal BB and NB coherence, we compared those latencies using a $t$ test with unequal variance assumption (Supplementary Fig. 4c). We computed the latency for each pair based on the median of the distribution of earliest times for which the coherence exceeds a given significant level (black contours in Supplementary Fig. 4a, b).

**Comparison of frequency distributions between spinal and cortical NB coherence.** For comparison of frequency distributions between the spinal and cortical NB coherence patterns, the normalized (Z-transformed) differences between the two patterns of coherence were tested using the nonparametric Monte Carlo method[59]. The Z-transformation was undertaken as:

$$Z = \frac{\left(atanh(|C_1(f)|) - \frac{1}{DF_1-2}\right) - \left(atanh(|C_2(f)|) - \frac{1}{DF_2-2}\right)}{\sqrt{\left(\frac{1}{DF_1-2}\right) + \left(\frac{1}{DF_2-2}\right)}},\qquad(6)$$

where $|C_1(f)|$ and $|C_2(f)|$ stand for each coherence, and $DF_1$ and $DF_2$ for the degrees of freedom ($2 \times$ (pairs) × (frequency bin)). Under the null hypothesis the two coherence frequency distributions are equal, the underlying datasets ((88 pairs against 127 pairs) × (64 frequency bins)) for two coherence patterns were randomly permuted, averaged, and calculated into the Z-scores above. The procedure was iterated 10,000 times to obtain an empirical distribution, from which 97.5th and 2.5th percentiles were assigned for upper and lower limits to determine a significance (Fig. 4b). Circular statistics on the phase distributions, such as Rayleigh test and Mardia–Watson–Wheeler test), were performed with Circstac toolbox[60].

**Spectral analysis on the multivariate autoregressive (MVAR) model.** Given the significant coherence found between LFPs and EMGs in the grip or hold phase, we further sought to examine whether the coherence reflects putatively causal interactions with a particular direction. Analyses of the causal interaction in the network can involve estimating the extent to which one signal influences another, and assessing whether the lags between them based on the measure are (physiologically) plausible.

For this purpose, we performed spectral analysis on the multivariate autoregressive (MVAR) model[61], that was estimated from the LFP and EMG time series in the same time window as used for standard coherence, 512 ms (128 points). The segmented signals were fitted to an MVAR model of two time series (ARfit package[62]) as described by the equation below:

$$\begin{bmatrix} y_1(n) \\ y_2(n) \end{bmatrix} = \sum_{k=1}^{p} A_k \begin{bmatrix} y_1(n-k) \\ y_2(n-k) \end{bmatrix} + \begin{bmatrix} \epsilon_1(n) \\ \epsilon_2(n) \end{bmatrix},\qquad(7)$$

where $y_1(n)$ and $y_2(n)$ are the two time series. The off-diagonal components of the 2-by-2 matrix $A_k$ predict the current sample ($n$) of $y_1$ and $y_2$ from the $k$th past sample of $y_1$ and $y_2$. The model order $p$ defines the maximum lag used to quantify such interactions. When the prediction error $\epsilon$ is minimized in the fitting of the coefficients of $A_k$, if the variance of $\epsilon_1$ is reduced by including the $y_2$ terms in the first equation in (7), then based on Granger causality, one can state that $y_2$ causes $y_1$, and vice versa.

The spectral representation of the MVAR process is derived considering the Fourier transformation of the equation above (7):

$$A(f)\begin{bmatrix} y_1(f) \\ y_2(f) \end{bmatrix} = \begin{bmatrix} \epsilon_1(f) \\ \epsilon_2(f) \end{bmatrix},\qquad(8)$$

where $A(f)$ is 2-by-2 coefficient matrix calculated as:

$$A(f) = \sum_{k=1}^{p} A_k e^{-i2\pi fkT},\qquad(9)$$

where $i$ is an imaginary unit, and $T$ is the sampling interval. Equation (8) is rewritten as:

$$\begin{bmatrix} y_1(f) \\ y_2(f) \end{bmatrix} = H(f)\begin{bmatrix} \epsilon_1(f) \\ \epsilon_2(f) \end{bmatrix},\qquad(10)$$

where $H(f)$ is 2-by-2 transfer function matrix calculated with $A(f)$:

$$H(f) = [I - A(f)]^{-1} = \overline{A}(f),\qquad(11)$$

where $I$ is the identity matrix.

Considering the trade-off between sufficient spectral resolution and overparameterization, we determined the model order as a value of 15 (60 ms for our 250 Hz sampling rate), as a comparable value of 10 (50 ms for 200 Hz sampling rate) was used in the previous study that focused on an MVAR model of sensorimotor cortical networks underlying the beta oscillation[63].

Directed coherence and partial directed coherence: We then derived two measures based on the transfer function matrix $H(f)$ or the coefficient matrix $A(f)$ in the MVAR spectral model, one called directed coherence[29], while the other called partial directed coherence[35].

Directed coherence ($\gamma_{ij}(f)$) is calculated as:

$$\gamma_{i \leftarrow j}(f) = |H_{ij}(f)|^2 \frac{S_{jj}(f)}{S_{ii}(f)}, \tag{12}$$

where $S_{kk}(f)$ is the power spectral density of the signal $k$, calculated based on the AR model as:

$$S(f) = H(f)VH(f)^{\mathrm{H}}, \tag{13}$$

where $V$ is the covariance matrix of the error term $\epsilon(f)$ and the superscript "H" denotes the Hermitian conjugate.

Partial directed coherence ($\pi_{ij}(f)$) is calculated as follows:

$$\pi_{i \leftarrow j}(f) = \frac{\frac{1}{\delta_i^2}|\overline{A}_{ij}(f)|^2}{\sum_{m=1}^{M} \frac{1}{\delta_m^2}|\overline{A}(f)_{mj}|^2}, \tag{14}$$

where $\delta_k$ represents a variance of $u_k$.

Partial directed coherence ($\pi_{ij}(f)$), reflecting the off-diagonal elements of $A(f)$, is nonzero if and only if direct causality from $y_j$ to $y_i$ exists, whereas directed coherence ($\gamma_{ij}(f)$), based on $H(f)$ that contains a sum of terms related to every transfer paths, is nonzero whenever any path connecting $y_j$ to $y_i$ is significant, reflecting both direct and indirect causality between $y_j$ and $y_i$. The two measures also differ in normalization; $\gamma_{ij}(f)$ is normalized with respect to the structure that receives the signal, whereas $\pi_{ij}(f)$ is normalized with respect to the structure that sends the signal.

As such, directed coherence provides a total causal influence as the amount of signal power transferred from one process to another but cannot distinguish direct causal effects from indirect ones. Conversely, partial directed coherence clearly measures the underlying interaction structure as it provides a one-to-one representation of direct causality, but is hardly useful as a quantitative measure because its magnitude quantifies the information outflow, which does not provide precisely how much information reaches downstream.

Documented directed coherence measures applied to corticomuscular interactions[29,31,32,64] have limited accuracy in estimating the phase–lag relationship, due to their inability to distinguish the direct and indirect causal effects. This is probably because the sensorimotor corticomuscular interaction comprises closed loops, including a bidirectional interaction between the motor cortex and the muscles[33,34]. In a nested loop, an oscillation in the past cycle would be recurrently summed, thereby leading to a phase shift owing to the synthesis of oscillations separated with a fixed time lag of the loop cycle. The degree of phase shift is variable for different frequency bands; when a lag in the time domain is converted to a phase in the frequency domain, the phase is increased as a function of the frequency band. Hence, an estimate of a lag based on a phase−lag plot tends to be shorter than actual transmission, as demonstrated by Campfens et al.[34]. The authors further showed in the simulation that partial directed coherence provided the most accurate estimate of the lag than any other directed coherence measures ever attained. This is a clear indication that partial directed coherence reflects a direct causal relationship between the two variables as theoretically explained above.

Considering the complementary properties of directed coherence and partial directed coherence measures, we decided to employ directed coherence to determine a causal influence between the cortex and muscles, and partial directed coherence to measure the phase–lag relationship for the causally defined interaction by directed coherence. By combining these two measures, we can reliably determine causal influences between the cortex and the muscles, and can also estimate the lag accurately in the presence of open or closed loops interposed in between.

Baker et al.[29] showed that the significance limit of directed coherence was comparable with that for standard coherence as stated in Eq. (5). Based on this assumption, we set the $\alpha$ level at the same value for the coherence analysis as for directed coherence ($p = 0.005$). For statistical tests on combined coherence across multiple recorded pairs (pooled coherence), we chose a nonparametric method according to Baker et al.[27] wherein we simply counted the percentage of bins at a particular frequency that exceeded the significance limit in the individual coherence spectra. The percentage was calculated by dividing the significant number by the total number of pairs at a given frequency; the significance limit was determined using binomial parameter estimation ($p = 0.005$) with the total number of pairs.

The phase of partial directed coherence for the significant bins determined by directed coherence was calculated as follows:

$$\theta(f) = \arg\left(\sum_{i=1}^{L} X_i^*(f)Y_i(f)\right), \tag{15}$$

where $L$ is the number of the data sections, $X$ and $Y$ denote each time series, and * denotes a complex conjugate. The 95% confidence limits on the phase estimates, ($\theta \pm \theta$) were determined according to[57]:

$$\Delta\theta(f) = 1.96\sqrt{\frac{1}{2L}\left(\frac{1}{\text{Coherence}(f)} - 1\right)}. \tag{16}$$

To determine if there is a fixed time delay between the two time series, we fitted a regression line to the phase–frequency relationship, as two correlated signals with a fixed time delay in the time domain give a linear function of frequency in the spectral domain. If the slope is significantly different from zero $p < 0.05$, $t$ test on the regression coefficient, the constant delay ($\tau$ ms) was estimated from the line's slope as follows:

$$\tau = -\frac{1000}{2\pi}A, \tag{17}$$

where $A$ is the line's slope (rad/Hz). A negative slope (positive delay) indicates that LFP leads EMG with a constant delay and vice versa.

**Statistics and reproducibility**. To recapitulate, LFP–EMG pairs with >99 trials were employed for analyses to ensure the reliability. A significance of the coherence for each frequency band was defined if the value exceeded a significance threshold level S (Eq. 5), with a level at $p = 0.005$. Dissociation of spinomuscular NB and BB coherence was performed using E–M algorithm on the distribution of integral of contours under GMM assumption; thereafter outliers in NB were determined using Smirnov–Grubbs test ($p < 0.05$). The difference in the onset latency between spinomuscular NB and BB coherence was examined using a $t$ test with unequal variance assumption. For comparison of frequency distribution between spinal and cortical NB coherence, the $Z$-transformed differences between the two patterns of coherence were tested using the nonparametric Monte Carlo method. The procedure was iterated 10,000 times to obtain an empirical distribution, from which 97.5th and 2.5th percentiles were assigned for upper and lower limits to determine a significance. To examine nonuniformly distribution or phase angle difference in spinal and cortical NB coherence, circular statistics such as Rayleigh test and Mardia–Watson–Wheeler test) were performed on the phase distributions of spinal and cortical NB coherence. All statistical measures were computed using MATLAB (MathWorks, Natick, MA, USA). Data were collected from four non-human primates, which performed the same task. For spinal or cortical recordings, data were sampled from at least two animals (Supplementary Fig. 3).

**Reporting summary**. Further information on experimental design is available in the Nature Research Reporting Summary linked to this article.

## Data availability

The data that support the findings of this study are available from the authors upon reasonable request. All source data underlying the graphs and charts presented in the main figures are available as Supplementary Data 1.

## Code availability

All custom codes will be available upon request. Please contact the corresponding author. MVAR analyses were performed with ARFit Toolbox (https://github.com/tapios/arfit). Circular statistics on phase distributions were performed with CircStat Toolbox (https://www.jstatsoft.org/article/view/v031i10).

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

## Acknowledgements

The authors would like to thank C. Sasaki, N. Takahashi and K. Isa for technical assistance. This work was supported by JSPS Grants-in-Aid (KAKENHI) Grant Numbers: 18020030, 18047027, 20020029, 20033025, 23300143, 26120003, 26250013 (to K.S.), 06J02928, 21700437, 23700482, 19H03975, 19H05311 (to T.T.), 10J05147 and 18K10984 (to T.O.). K.S. was funded by the JST Precursory Research for Embryonic Science and Technology Program.

## Author contributions

T.T. and K.S. conceived the idea and designed the experiments. T.T., T.O., and K.S. performed experiments and T.T. and T.O. performed analyses of the data. T.O., T.T. and K.S. discussed the results and prepared the manuscript.

## Competing interests

The authors declare no competing interests.
