## [Peer Review File · Communications Biology]

Reviewers' comments:

Reviewer #1 (Remarks to the Author):

This manuscript is an analysis of the coherence between both cortical and spinal LFP, and EMG activity measured from arm and hand muscles during precision grip. The authors argue their results are evidence within the beta band, of parallel control loops mediating the initial phasic grasp part of the behavior (via a spinal, reflexive loop), and the latter hold part (via cortex). The difference in the grasp and hold phases of the spinal and cortical NB coherence is striking. Although there is strong prior evidence of hold-period coherence from cortex, the spinal coherence during grasp is novel. To my understanding, the origin and function of beta band oscillations and coherence remains uncertain. This study may help to establish its relation to a precision grip, which has been extensively studied at the single unit level in monkeys for several decades. However, I'd be more convinced of the conclusions drawn here, if they can be related more directly to those classical observations.

Why the association of cortex with the tonic phase, when virtually all single-unit results find the opposite result, that M1 activity is largely phasic? Although there is indeed some evidence for increased tonic activity in CM cells, and those located more caudally in M1, even these cells remain significantly more phasic than spinal interneurons or muscles. Were the signals in the current study recorded predominantly from these potentially more tonic sources? Was there any difference between LFP recorded from rostral and caudal sites in cortex? There is some evidence of integrator-like activity in the cord. This too, seems opposite the current result. The cortico and spinomuscular lags and total loop times reported here are the same. How are we to account for the corticospinal delays?

Detailed comments

24: "During a dynamic movement, the trajectory and speed are often required to be controlled, whereas holding-still requires the stability of the muscle force."

Of course, one could also argue that holding still is simply controlling speed=0, with well-controlled muscle force in both cases.

1D: This figure is a bit confusing. The "coherence" reference in the titles is only to the lowest set of panels. The tiny little, ultra narrow-band coherence in ii and iii is a bit unconvincing. The examples in F2 are a better. The white trace is EMG power. Why doesn't this better reflect the EMG spectrogram in the middle row? The disparity is particularly striking in iii.

105: "This dissociation was largely consistent with the distributions in frequency width (Figure S1B-i, vertical histogram)"

Really? What is R2? It looks like <0.5 to me. Please clarify what each point corresponds to. A single recording site from a single session? Which muscles?

107: "For corticomuscular coherence we saw solely unimodal distributions in both frequency width and contour integrals (Figure S1B-ii)..."

There seems to be virtually no correlation between the two measures. I'm not sure that matters, but it's worth a comment. The different x-axis lengths in i and ii obscure the difference between the two distributions. I'd make them the same, unless there is some reason why this is not appropriate.

130: "The latency of the spinal broad-band coherence indicates that the onset occurred prior to grip onset, as well as spinal beta-band coherence onset by c.a. 100 ms."

How much does this latency difference have to do simply with the great bandwidth (faster risetime) of the BB signal?

Fig 3: Why is there so little coherence from any of these sources with muscles innervated by the radial nerve? Were these muscles active in the task? How well does EMG magnitude predict coherence?

171: "It is notable that the aggregate median ... delay was... a reciprocal of the central frequency of the coherence..."

Just what makes this "notable"? Presumably it is not inevitable, or it would not be notable. What does make it notable? Why is it important? Is it, in fact, inevitable and simply a sanity check on the lag measurements themselves?

195: "Here, we found broad-band coherent patterns closely resemble typical temporal EMG profiles..."
In Fig 1D it looks a lot more like LFP than EMG.

307: "Conceptually, the separation of the circuitry for each phase appears common to the neural circuit implementation for saccadic eye movements accommodated in the brainstem"
This statement seems to suggest that the separate, parallel architecture proposed here is similar to that of the eye movement system. It clearly is not, as that is modeled as a serial pulse generator followed by integrator. The subsequent few sentences clarifies the authors' perspective, but I think this sentence needs to be changed.

Minor things

39: "_albeit_ the individual premotoneuronal cells could affect the target motoneuron activities
Wrong word choice." "Although"?

48: "it is uncertain for the separation of neural circuitry underlying dynamic and static skeletomotor control..."

I am not sure what uncertainty this sentence refers to.

64: "static hold controls"

It is unclear to me whether this is a reference to an experimental control condition (it doesn't seem to be) or to the need for the monkey to maintain a controlled grip force. Please clarify.

112: "showing significant coherence during the grip and hold phases, respectively"
showing significant coherence during the grip or hold phases, respectively, either alone or in combination.

139: "We also found each phase distribution clustered to a specific angle"

I don't find the phase angles to be particularly intuitive. What lag do they correspond to?

145: "between an LFP and multiple EMGs at a given recording site"

I assume this means, "between an LFP at a given recording site and multiple EMGs"

184: "the trans-spinal loop involves the prime movers of the precision grip muscles with interactions among local forearm flexors, whereas the trans-cortical feedback loop arise largely through the recruited finger muscles."

What is meant to be the distinction between, "prime movers of the precision grip muscles" and "finger muscles"?

367: "For spinal recordings, we implanted a recording chamber on the cervical vertebra (C4-C7) of monkeys U, A, E and S"

Did "S" fail for some reason?

397: "EMGs were high-pass filtered at 30 Hz (4th-order Butterworth filter), rectified, and down-sampled to 250 Hz."

Was there a low-pass stage between rectification and down-sampling?

416: "or an onset of hold release"

There must be a typo here.

525: "provided the most accurate estimate of the lag that any other directed coherence measures never attained."

Another typo?

Reviewer #2 (Remarks to the Author):

This manuscript by Oya and colleagues describes a study regarding the neuronal mechanisms of volitional limb motor control. The main question they asked is whether the brain deploys a distinct neuronal circuit or a common one for phasic and static motor activity using the same set of muscles. Although beta-band oscillations in the motor system, particularly in the motor cortex and their relationship between muscle activity have been studied extensively (Baker et al. 2007), 1) beta-band coherent oscillations of neuronal structures with muscles have not been reported during the phasic phase (when the fingers are moving), and 2) the origin of the coherent oscillations remains unraveled. Using macaque monkey as the experimental model, they recorded the local field potential (LFP) from the motor cortex and the spinal cord as well as electromyogram (EMG) of the forelimb muscles during a precision grip task in which contained a distinct phasic/static motor activity phase. They then analyzed the beta-range coherence of the LFPs and EMGs.

The main findings:

- 1) The beta-range coherence with EMGs at spinal cord and motor cortex emerges mainly during gripping (phasic) action with the former and with the latter during the holding (static) action. The beta-range coherence of the EMG and a neuronal structure during the phasic phase has not been described in previous studies while the latter has been demonstrated before (see Baker et al. 2007).
- 2) By directional information analyses they show that these beta-band coherences are likely to be a result of the reciprocal interaction (feedback loop) between the neuronal structure and sensory afferents.
- 3) They show that these two feedback loops differ in the muscles involved (i.e., spinal local feedback vs. cortical divergent feedback loops).

From these results the authors argue that dedicated feedback circuits comprising spinal and cortical structures underlie dynamic and static controls of dexterous hand actions.

The manuscript is well-written and the data are clearly presented. The challenging experiments, especially simultaneously recording the activity of forelimb/hand muscles together with the neuronal activity of the cortex and the spinal cord in a behaving monkey were elegantly designed, and analyses were carried out in a sound manner.

The results highlight the importance of the lower motor structures including the spinal cord and sensory afferents during volitional movement which has been historically overlooked. They are in line with previous works including studies by the authors (listed in their reference list). They may provide new ideas for the larger neuroscience field including clinical neurology and motor rehabilitation. I have several comments.

1. My main concern is the lack of direct evidence of the functional significance of the proposed feedback loop to justify the conclusion. While the results do indicate that there are distinct (but possibly overlapping) networks controlling different motion using the same set of muscles, the identity of neurons and their relevant connections remain unknown. A new set of challenging experiments such as directly recording/controlling neurons, are required to investigate this point which is probably out of the scope of the current study.
2. Related to point 1, I think the manuscript gives an impression that the appearance of the beta-band coherence represents how active the network is. However, in this study, it could merely reflect the different amount of feedback signal (sensory input) itself during the dynamic and static phase which is not necessary a central mechanism.
3. In one of their previous studies (Takei and Seki 2008), they showed that the majority of the spinal premotor interneurons exhibited both phasic and tonic activities simultaneously and suggested that these neurons integrate dynamic and static motor commands to produce the final motor output. How would the distinct feedback loops described in the present study may contribute to this integration? A general point has been mentioned in the discussion (lines 258-261), but it could be also discussed in this context.
4. Lines 110-117: While I agree that the majority of the cortical narrow-band coherence pairs seems to be significant during the hold phase (Figure 2A-iii), 20% of them are significant during the grip phase and these pairs should not be overlooked. Do they reflect the efferent component observed in Fig 6iii (From Fig 7 it seems unclear)? I think this point could be added to the discussion.
5. It would help the readers outside the motor network field to show a representative recording trace of the LFPs and EMGs of all the muscles recorded during the task in the supplemental figure.
6. M1 neurons have been shown to encode reward signals (e.g. Ramakrishnan et al, PNAS 114 (24) E4841-E4850, 2017). Since the performance of the task is driven by the reward, could they be affecting the cortical activity of the holding phase closer to the reward?

Reviewer #3 (Remarks to the Author):

Oya et al.,

Brief summary of the manuscript

Oya et al. address the question of whether dynamic and static forelimb movements are controlled through separate or shared neural circuits. They did this by recording simultaneously from different neural motor control centers (primary motor cortex and spinal cord) and forelimb muscles in monkeys while they performed a forelimb movement comprising dynamic and static phases separated in time (precision grip). Coherence analysis between local field potentials (LFPs) recorded from neural centers and muscles during the different movement phases revealed: (1) spinal cord and muscles to be coherent across a broad-band of frequencies during both the dynamic and static phases of movement (broad-band coherence); (2) spinal cord and muscles to be coherent at beta-band frequencies (15-30Hz) during the dynamic phase of movement; and (3) motor cortex and muscles to be coherent at beta-band frequencies during the static phase of movement. They also showed differences between these 3 types of coherence in their latencies relative to movement, their muscle distributions, and their directions of causality. Overall the data support the notion that different spinal and cortical neural networks underlie dynamic and static control of forelimb movements.

Overall impression of the work

These experiments are at the extreme end of difficulty and the Authors should be congratulated for completing and disseminating their work. The research directly addresses several important questions in neural control of movement, it employs sound methods, and it provides concise and novel data relevant to each question with relatively clear outcomes. I support publication in Communications Biology.

Specific comments

1. Parts of the Introduction have sentences that are too long, leading to a lack of clarity.

In particular the consecutive sentences beginning line 33 and ending 42 are both unclear to me. The first sentence I think you mean "Neurons in caudal motor cortex and those with direct connections to spinal motoneurons (e.g. corticomotoneuronal and premotor spinal interneurons) tend to discharge tonically during the static phase." The point of the second sentence is less clear to me. It seems contradictory since the Authors seem to be saying that the cited studies are deficient in that the effective connectivity of recorded cells to muscles was not assessed, however it was assessed and found to be incongruent with their discharge properties.

Also, the sentence beginning line 45 is long and difficult to follow. Are points 1 and 2 separate questions that should be communicated as such?

2. Table S2. The vast majority of data pertaining to spinal cord LFP and EMG pairs were harvested from 1 of 4 monkeys (monkey E). Despite this I could find no mention anywhere in the manuscript of how well spinal cord LFP and EMG data from the other 3 monkeys matched monkey E. Such a statement is necessary to assess how generalizable the conclusions drawn from the study are across monkeys.

Similarly, although the distribution of data pertaining to motor cortex LFP and EMG pairs is better balanced across the 2 monkeys, a statement about how similar data from each monkey are would be helpful.

3. The Authors rightly point out in the Introduction that neurons in different parts of the motor cortex and spinal cord exhibit discharge patterns better correlated with specific phases of movement (dynamic versus static). The manuscript would be improved if similar analyses were applied to the Authors' data. This would inform upon questions such as: (1) are there any differences between rostral and caudal motor cortex? And (2) are broad-band spinal locations, which are suggested to be more efferently connected with muscles, located more ventrally than narrow-band spinal locations, which are suggested to be more bidirectionally (efferently and afferently) connected?

Also, the cortical recording chambers were located over the pre- and post-central gyri. Is there any data from LFP recordings in post-central gyrus? This may help support and develop the conclusions drawn.

Responses to comments:

We are very grateful to the careful review of our manuscript and the constructive comments provided. We have incorporated reviewers' suggestions into the manuscript and have added some new analyses to address the issues raised by the reviewers (e.g., on the relationship of recording sites with corticomuscular coherence patterns in the motor cortex, and depth profiles of spinomuscular coherence patterns). Furthermore, we have added new supplemental figures concomitantly: Figure S3 for depth profiles of spinomuscular coherence, Figure S4 for corticomuscular coherence patterns with respect to recording sites in the motor cortex, Figure S5 for example raw traces for EMGs and LFPs, and Figure S6 for pie charts showing the proportions of significant coherence from individual animals. In addition, we have had our manuscript edited by a professional editor to improve clarity of the text. Although we have included a red-lined version of the manuscript for your reference, it may not be readable because of extensive revision. Therefore, we here submit the plain text as a main manuscript file.

We feel that the revised manuscript is a suitable response to the comments, and is significantly improved from the version previously submitted. We believe that it is now suitable for publication in *Communications Biology*.

We have prepared the revised manuscript in two versions. In "main.pdf", added text is colored in blue. To show the discarded text clearly, we have prepared "main_markup.pdf", in which added text is underlined and colored in blue, and discarded text is struck out and colored in red.

Below are our responses (blue) to the reviewers' comments (black):

Referee expertise:

Referee #1: Motor cortex

Referee #2: Motor control and behavior

Referee #3: Behavioral neuroscience

Reviewers' comments:

Reviewer #1 (Remarks to the Author):

This manuscript is an analysis of the coherence between both cortical and spinal LFP, and EMG activity measured from arm and hand muscles during precision grip. The authors argue their results are evidence within the beta band, of parallel control loops mediating the initial phasic grasp part of the behavior (via a spinal, reflexive loop), and the latter hold part (via cortex). The difference in the grasp and hold phases of the spinal and cortical NB coherence is striking. Although there is strong prior evidence of hold-period coherence from cortex, the spinal coherence during grasp is novel. To my understanding, the origin and function of beta band oscillations and coherence remains uncertain. This study may help to establish its relation to a precision grip, which has been extensively studied at the single unit level in monkeys for several decades. However, I'd be more convinced of the conclusions drawn here, if they can be related more directly to those classical observations.

Why the association of cortex with the tonic phase, when virtually all single-unit results find the opposite result, that M1 activity is largely phasic? Although there is indeed some evidence for increased tonic activity in CM cells, and those located more caudally in M1, even these cells remain significantly more phasic than spinal interneurons or muscles. Were the signals in the current study recorded predominantly from these potentially more tonic sources? Was there any difference between LFP recorded from rostral and caudal sites in cortex? There is some evidence of integrator-like activity in the cord. This too, seems opposite the current result. The cortico and spinomuscular lags and total loop times reported here are the same. How are we to account for the corticospinal delays?

We are very grateful to the reviewer for appreciating the novelty and significance of our study. We also thank the review for insightful points. We address the first and second questions: "*Why the association of cortex with the tonic phase, when virtually all single-unit results find the opposite result, that M1 activity is largely phasic?*" and "*Were the signals in the current study recorded predominantly from these potentially more tonic sources?*" here.

As the reviewer points out, the discharge patterns of most of the M1 neurons are phasic-tonic, whether they are of the rostral or caudal part (Maier et al., 1993), which seems incongruent with the pattern of corticomuscular coherence that is evident during

the static hold phase. However, it is stressed that the LFPs do not reflect the discharge rate of individual neurons (Murthy and Fetz, 1996), but reflect the synchronized discharges of many neurons (Denker et al. 2011), which are known to act differentially from rate modulation (Riehle et al., 1997). Neuronal synchrony between the neurons in the motor cortex is not evident during the dynamic phase, despite increased discharges, whereas it is pronounced during the static phase (Baker et al., 2001). Consistently, the corticomuscular coherence manifests primarily during the static phase, irrespective of the part (rostral or caudal) (Baker et al., 1997). To the best of our knowledge, the previous literature points to the presence of corticomuscular coherence in the static phase.

Therefore, assuming that the coherence pattern itself would be invariable in the motor cortex, we made recordings mainly from the caudal part, for better yields (Baker et al., 1997, Supplemental figure S4). As such, we cannot provide a direct comparison between the subdivisions. The cited comparison between rostral vs. caudal parts was not intended for further exploration, but to suggest a limitation of discharge rate modulation for exploration of dynamic and static control. We admit that the sentence was unclear, and hence we have revised the sentence in lines 32-46.

We next address the question: *“There is some evidence of integrator-like activity in the cord. This too, seems opposite the current result”*.

We have assumed the “integrator-like activity in the cord” to be observations of a phasic-tonic discharge pattern of spinal neurons (e.g., in Takei & Seki, 2013). As mentioned above, previous studies have shown that cortical LFPs, as reflected by synchronization of neurons, are not correlated with discharge rates of individual neurons (Murthy and Fetz, 1996). Rather, neuronal discharge and synchronization supposedly act cooperatively to achieve an efficient neuronal computation (Riehle et al., 1997; Womelsdorf et al., 2007). In the motor cortex, it is proposed that enhanced efficacy through synchronized oscillation may contribute to saving discharge rates (Baker et al., 1999). In contrast, in the spinal cord, an active decorrelation mechanism within IN networks appears to work during the static phase; while spinal interneurons increase the firing rates, the synchronization is reduced during the active hold (Prut and Perlmutter, 2003). This decorrelation mechanism probably acts to maintain information capacity of IN networks by preventing excessive motoneuron synchrony. The absence

of spinomuscular coherence during the hold period may be related to this decorrelated mechanism. As such, in the process of integration of final motor output, the dynamic phase neuronal discharge in the cord may be enhanced via feedback-induced synchronization. Conversely, in the static phase, the neuronal discharges in the cord are actively decorrelated to maintain motoneuron synchrony at functionally appropriate levels, whereas cortical descending drive utilizes the feedback-related synchronization to efficient motor output. We added this discussion in lines 270-287.

We then address the question: “*The cortico and spinomuscular lags and total loop times reported here are the same. How are we to account for the corticospinal delays?*”

The fastest putative latency for responses to the peripheral electrical stimulation via the trans-spinal and transcortical loops could be calculated as a sum of afferent and efferent conduction times. The latencies of spinal unit responses to electrical stimulation were about 4–7ms (Confais et al., 2017), and that of muscles to intraspinal microstimulation (ISMS) were 5–6 ms (Takei and Seki, 2010). Thus, the total latency for the trans-spinal loop could be around 9–13ms. Likewise, latencies of cortical response to electrical stimulation were around 10–11 ms (Seki and Fetz, 2012), and that of muscles to ICMS were 6–8 ms (Fetz and Cheney 1980), resulting in a total latency of 16–19 ms. The time lag calculated from the coherence analyses, however, does not necessarily agree with the fastest delays, because there are multiple routes for computational processes with various conducting fibers in which the signal is communicated. As the coherence captures a population activity, which reflects various neuronal communications taking different routes to the spinal motoneurons, one could expect to see the average of taken routes as a best fit for LFP-EMG time delay. The apparent discrepancies for the lags, therefore, would be interpreted as mixture of temporally dispersed volleys and processings through the routes. Taking these factors into consideration, one could assume that the difference in fastest latencies (ca. 3–10 ms) was not dissociable between the lag estimates via trans-spinal and transcortical loops.

Detailed comments

24: “During a dynamic movement, the trajectory and speed are often required to be controlled, whereas holding-still requires the stability of the muscle force.”

Of course, one could also argue that holding still is simply controlling speed=0, with

well-controlled muscle force in both cases.

We thank for the thoughtful comment. As the reviewer has suggested, dynamic and static control can be achieved via a unified control process, and therefore it is not trivial to assume that different processes are underlying. We agree with the reviewer on this point, and have revised the sentences in lines 23-31 as follows:

“A long-standing fundamental question is whether the dynamic and static actions are achieved through similar or specialized control processes. Similar control processes for each action has the advantage of generality and conserved neural circuitry; specialized control processes have the advantages of context-dependency and flexibility. However, specialized processes require neural circuits for greater dedication and expertise (Kurtzer et al., 2005). Elucidation of the underlying mechanisms of control processes is the key to understanding a motor system that is confronted with conflicting demands between generality and flexibility. Specialized processes and underlying neural circuits have been demonstrated in the primate brainstem for the dynamic and static control of saccadic eye movement (Robinson, 1973; Shadmehr, 2017).”

1D: This figure is a bit confusing. The “coherence” reference in the titles is only to the lowest set of panels. The tiny little, ultra narrow-band coherence in ii and iii is a bit unconvincing. The examples in F2 are a better. The white trace is EMG power. Why doesn't this better reflect the EMG spectrogram in the middle row? The disparity is particularly striking in iii.

We thank for pointing out a lack of clarity. Figure 1D shows representative pairs, not averaged ones such as those shown in Figure 2. The significant bands for narrow-band coherence are comparable to the results obtained in other reports (Baker et al. 1997). The white traces stand for the averaged amplitudes for respective EMGs in the time domain. We agree with the reviewer that overlaying the time-domain signals on the frequency-domain signals is confusing, and therefore we have removed the white traces. We have also changed the scale of the EMG spectrogram and colormap for better illustration in i), Figure 1D. The scales and colormaps for other EMG spectrograms have also been changed.

105: “This dissociation was largely consistent with the distributions in frequency width (Figure S1B-i, vertical histogram)”

Really? What is R2? It looks like <0.5 to me. Please clarify what each point corresponds to. A single recording site from a single session? Which muscles?

107: “For corticomuscular coherence we saw solely unimodal distributions in both frequency width and contour integrals (Figure S1B-ii)...”

There seems to be virtually no correlation between the two measures. I’m not sure that matters, but it’s worth a comment. The different x-axis lengths in i and ii obscure the difference between the two distributions. I’d make them the same, unless there is some reason why this is not appropriate.

We thank for pointing out a lack of clarity. Responses to the comments for lines 105 and 107 (in the original manuscript) are addressed together here.

In Figure S1B, each point in the scatterplot represents a pair of significant coherence (i.e., single-site LFP and single muscle). In the scatterplot, we do not intend to show a correlation between the two measures, but show how well the points are distributed in the categorized quadrants by two measures, that is, significant integrated contour in wavelet coherence (vertical axis) vs. the bandwidth of normal coherence (horizontal axis). Since the former takes a temporal profile into consideration as well as the significance bandwidth, one could avoid misclassification of broad-band coherence reflecting motoneurons with low firing rates such as upper arm muscles, as shown in the lower-right quadrant. The misclassification by the integrated contour is relatively small, as shown in the upper-left quadrant. This measure also successfully dissociates two underlying Gaussian distributions and is therefore likely to be justified from a statistical point of view. The explanation of the figure has been revised in lines 109-113.

130: “The latency of the spinal broad-band coherence indicates that the onset occurred prior to grip onset, as well as spinal beta-band coherence onset by c.a. 100 ms.”

How much does this latency difference have to do simply with the great bandwidth (faster risetime) of the BB signal?

We thank for an insightful point. In the analysis, we took the median of the onset times for significant frequency bands, which were subject to subsequent comparison. We added a representative example in Figure S2 to better clarify the onset line in the contour and spectrogram. As shown in Fig 1D and the new Figure S2, there is no tendency in the lag toward the specific band. Rather, significant bands emerge at almost the same onset. In addition, by taking the median of the slightly variable onsets for the

respective band, we present unbiased values. We have added a description for the way the onset was calculated in lines 484-488.

Fig 3: Why is there so little coherence from any of these sources with muscles innervated by the radial nerve? Were these muscles active in the task? How well does EMG magnitude predict coherence?

The scarcity of coherence for radial nerve muscles, as the reviewer points out, is very interesting to us too. The radial-innervated muscles are indeed recruited in the task, as shown in Figure S5. Accordingly, presence of muscle activity is not directly related to the occurrence of coherence. As for the extent to which the muscle is activated, it is quite difficult to quantify the magnitude of an EMG across muscles, since there is no way to normalize the EMG amplitudes without taking maximal voluntary contractions, which is practically impossible for the animal to perform in the experimental setup.

171: “It is notable that the aggregate median ... delay was... a reciprocal of the central frequency of the coherence...”

Just what makes this “notable”? Presumably it is not inevitable, or it would not be notable. What does make it notable? Why is it important? Is it, in fact, inevitable and simply a sanity check on the lag measurements themselves?

We thank for pointing out a lack of clarity. In neural networks, coherence between oscillators is theoretically facilitated by reciprocal connections between network neurons (a neural loop) and/or common input or drive from an external source (Aumann and Prut, 2015). Theoretically, a way to help distinguish these possibilities is to examine phase-shifts between coherence oscillations in different parts of the network. So far, however, the entire lags for directed coherence frequency did not agree in the previous experimental studies. In that sense, we believe our findings are notable since our results indicate agreement for the first time. However, we think these accounts are redundant and less suitable for Results section, as they are discussed in Discussion section in lines 255-269. We therefore removed the accounts.

195: “Here, we found broad-band coherent patterns closely resemble typical temporal EMG profiles...”

In Fig 1D it looks a lot more like LFP than EMG.

We thank for pointing out a lack of clarity. In Figure1D, to avoid confusion, we have changed the scale of the EMG spectrograms. Also, we have changed the sentence to “*In the present work, we found BB coherent patterns representing common temporal profiles of both LFPs and EMGs (i.e., phasic-tonic activity)*” (lines 201-202).

307: “Conceptually, the separation of the circuitry for each phase appears common to the neural circuit implementation for saccadic eye movements accommodated in the brainstem”

This statement seems to suggest that the separate, parallel architecture proposed here is similar to that of the eye movement system. It clearly is not, as that is modeled as a serial pulse generator followed by integrator. The subsequent few sentences clarifies the authors’ perspective, but I think this sentence needs to be changed.

We thank for pointing out a lack of clarity. We have revised the sentence to “*Although specific implementation of neural circuitry is not the same as saccadic eye movements accommodated in the brainstem (Robinson, 1973; Jurgens, 1981; Shadmehr, 2017), separating the circuits for each phase may be common to both eye and limb motor circuits.*” (line 336-339).

Minor things

39: “_albeit_ the individual premotoneuronal cells could affect the target motoneuron activities

Wrong word choice.” “Although”?

The sentence was deleted in the course of revision.

48: “it is uncertain for the separation of neural circuitry underlying dynamic and static skeletomotor control...”

I am not sure what uncertainty this sentence refers to.

The paragraph, including this sentence, has been revised for better clarification (lines 47–57).

64: “static hold controls”

It is unclear to me whether this is a reference to an experimental control condition (it doesn’t seem to be) or to the need for the monkey to maintain a controlled grip force.

Please clarify.

The sentence refers to the latter. We have revised the sentence to clarify our intended meaning (line 67).

112: “showing significant coherence during the grip and hold phases, respectively”
showing significant coherence during the grip or hold phases, respectively, either alone or in combination.

The phrase has been revised: “and” has been replaced with “or” (line 118).

139: “We also found each phase distribution clustered to a specific angle”

I don’t find the phase angles to be particularly intuitive. What lag do they correspond to?

Phase lags are shown as red lines in Figure 4B i) and ii). We have specified them (line 149).

145: “between an LFP and multiple EMGs at a given recording site”

I assume this means, “between an LFP at a given recording site and multiple EMGs”

That is exactly we meant. We have revised the sentence accordingly (lines 154–155).

184: “the trans-spinal loop involves the prime movers of the precision grip muscles with interactions among local forearm flexors, whereas the trans-cortical feedback loop arise largely through the recruited finger muscles.”

What is meant to be the distinction between, “prime movers of the precision grip muscles” and “finger muscles”?

We thank for pointing out a lack of clarity. We meant that prime movers for the muscles mainly contribute to index finger and thumb contraction, as those muscles mechanically contribute to precision grip. For the sake of accuracy, we have changed the expression to index finger muscles (FDI and FDP_r) in lines 190–191.

367: “For spinal recordings, we implanted a recording chamber on the cervical vertebra (C4–C7) of monkeys U, A, E and S”

Did “S” fail for some reason?

For a technical reason, a recording chamber came off too early in monkey S to collect as

much data as other monkeys. Due to the scarcity of data, the coherent LFP and EMG pairs were not found, after eliminating the cross-talked EMGs.

397: “EMGs were high-pass filtered at 30 Hz (4th-order Butterworth filter), rectified, and down-sampled to 250 Hz.”

Was there a low-pass stage between rectification and down-sampling?

We thank for pointing out a lack of description. EMGs were high-pass filtered at 30 Hz (4th-order Butterworth filter), rectified, and down-sampled to 250 Hz by averaging every 20 bins. The (moving) averaging involves a low-pass filtering with the cut-off frequency being ca. 110Hz ($F_c = 0.443/M * F_s$; F_c : cut-off frequency, M : bin number, F_s : sampling frequency) (Smith, 1999). We have added these descriptions to lines 428–430.

416: “or an onset of hold release”

There must be a typo here.

We thank for pointing out a confusing account. We have revised “either an onset of grip or an onset of hold release” to “either an onset of grip or an onset of release” in lines 449 and 476.

525: “provided the most accurate estimate of the lag that any other directed coherence measures never attained.”

Another typo?

We thank for pointing out a confusing account. We have revised “never” to “ever” in line 561.

Reviewer #2 (Remarks to the Author):

This manuscript by Oya and colleagues describes a study regarding the neuronal mechanisms of volitional limb motor control. The main question they asked is whether the brain deploys a distinct neuronal circuit or a common one for phasic and static motor activity using the same set of muscles.

Although beta-band oscillations in the motor system, particularly in the motor cortex and their relationship between muscle activity have been studied extensively (Baker et

al. 2007), 1) beta-band coherent oscillations of neuronal structures with muscles have not been reported during the phasic phase (when the fingers are moving), and 2) the origin of the coherent oscillations remains unraveled.

Using macaque monkey as the experimental model, they recorded the local field potential (LFP) from the motor cortex and the spinal cord as well as electromyogram (EMG) of the forelimb muscles during a precision grip task in which contained a distinct phasic/static motor activity phase. They then analyzed the beta-range coherence of the LFPs and EMGs.

The main findings:

- 1) The beta-range coherence with EMGs at spinal cord and motor cortex emerges mainly during gripping (phasic) action with the former and with the latter during the holding (static) action. The beta-range coherence of the EMG and a neuronal structure during the phasic phase has not been described in previous studies while the latter has been demonstrated before (see Baker et al. 2007).
- 2) By directional information analyses they show that these beta-band coherences are likely to be a result of the reciprocal interaction (feedback loop) between the neuronal structure and sensory afferents.
- 3) They show that these two feedback loops differ in the muscles involved (i.e., spinal local feedback vs. cortical divergent feedback loops).

From these results the authors argue that dedicated feedback circuits comprising spinal and cortical structures underlie dynamic and static controls of dexterous hand actions.

The manuscript is well-written and the data are clearly presented. The challenging experiments, especially simultaneously recording the activity of forelimb/hand muscles together with the neuronal activity of the cortex and the spinal cord in a behaving monkey were elegantly designed, and analyses were carried out in a sound manner. The results highlight the importance of the lower motor structures including the spinal cord and sensory afferents during volitional movement which has been historically overlooked. They are in line with previous works including studies by the authors (listed in their reference list). They may provide new ideas for the larger neuroscience field including clinical neurology and motor rehabilitation. I have several comments.

1. My main concern is the lack of direct evidence of the functional significance of the

proposed feedback loop to justify the conclusion. While the results do indicate that there are distinct (but possibly overlapping) networks controlling different motion using the same set of muscles, the identity of neurons and their relevant connections remain unknown. A new set of challenging experiments such as directly recording/controlling neurons, are required to investigate this point which is probably out of the scope of the current study.

We appreciate the careful review of the manuscript and thank the reviewer for appreciating the significance of our study, as well as thoughtful comments. We recognize the importance of clarifying the identities of neurons participating in the networks as more detailed connections at the cell level, as suggested by the reviewer. At the same time, as the reviewer also suggested, that point is out of the scope of this study. We believe, however, that the current results facilitate further exploration for the functional separation of neural circuitry, including the issues pointed out, with a focus on sub-cortical structures during voluntary motor control. This caveat has been added to sentences in lines 289-292 and 363-364.

2. Related to point 1, I think the manuscript gives an impression that the appearance of the beta-band coherence represents how active the network is. However, in this study, it could merely reflect the different amount of feedback signal (sensory input) itself during the dynamic and static phase which is not necessary a central mechanism.

We thank for providing a critical view on the interpretation. The reviewer states a view that the coherence merely reflects the different amounts of feedback signal (sensory input) itself, and this is not necessarily attributable to a central mechanism. To us, the definition of the amount of feedback signal (sensory input) is somewhat unclear. If that was used to mean the extent to which the activity of afferent fibers is reflected, e.g., amplitude of sensory-evoked potentials (SEPs), certainly SEPs observed in the spinal cord is larger than that in the motor cortex (Seki and Fetz, 2012). This statement, still, cannot explain the observations reported in the current study. First, it is not plausible to assume that the amount of sensory feedback is altered depending on dynamic or static phases without a central mechanism. Put another way, without some filtering (gating) mechanism, a comparative degree of coherence would be present throughout the grip and hold phases, but this is not the case; spinomuscular coherence diminishes during the static phase, whereas corticomuscular coherence emerges in the static phases. Second, a

mere feedback signal cannot account for significant directed coherence from LFPs (spinal cord and/or motor cortex) to EMGs. Third, related to the second issue, the bidirectional interaction, with a reasonable latency, gives corroboration that the coherence is generated via a loop. These observations cannot be made if it is caused by mere sensory feedback. Collectively, the coherence is quite likely to reflect a network activity that involves the motor cortex, spinal cord, and the muscles.

3. In one of their previous studies (Takei and Seki 2008), they showed that the majority of the spinal premotor interneurons exhibited both phasic and tonic activities simultaneously and suggested that these neurons integrate dynamic and static motor commands to produce the final motor output. How would the distinct feedback loops described in the present study may contribute to this integration? A general point has been mentioned in the discussion (lines 258-261), but it could be also discussed in this context.

We thank for insightful comments. We consider the issue as quite similar to the point raised by reviewer 1 in the comment “*There is some evidence of integrator-like activity in the cord. This too, seems opposite the current result*”.

Please see our response to the comment.

4. Lines 110-117: While I agree that the majority of the cortical narrow-band coherence pairs seems to be significant during the hold phase (Figure 2A-iii), 20% of them are significant during the grip phase and these pairs should not be overlooked. Do they reflect the efferent component observed in Fig 6iii (From Fig 7 it seems unclear)? I think this point could be added to the discussion.

We appreciate a thoughtful point made by the reviewer. To note and discuss the presence of corticomuscular coherence found during the dynamic phase, we have added these remarks to lines 247–250: “*A small but statistically significant fraction of pairs (5.5 %) represents efferent components during the dynamic phase (Figure 6A-iii)), with the averaged onset being 0.06 ± 0.14 s with respect to grip onsets. These may reflect some contribution of synchronized cortical output to dynamic control. It remains unknown how the corticomuscular coherence emerges*”.

5. It would help the readers outside the motor network field to show a representative

recording trace of the LFPs and EMGs of all the muscles recorded during the task in the supplemental figure.

We thank for a helpful comment. A figure showing raw data traces for EMG, LFP, and finger force has been added as supplemental figure S5.

6. M1 neurons have been shown to encode reward signals (e.g. Ramakrishnan et al, PNAS 114 (24) E4841-E4850, 2017). Since the performance of the task is driven by the reward, could they be affecting the cortical activity of the holding phase closer to the reward?

We thank for making an important technical point. In this experiment, the reward was given more than 1s after the release, and more 1s was given before the “go” cue. Thus, the influence of reward-related activity is negligible for the current observations. We have added the timing of the reward to line 388.

Reviewer #3 (Remarks to the Author):

Oya et al.,

Brief summary of the manuscript

Oya et al. address the question of whether dynamic and static forelimb movements are controlled through separate or shared neural circuits. They did this by recording simultaneously from different neural motor control centers (primary motor cortex and spinal cord) and forelimb muscles in monkeys while they performed a forelimb movement comprising dynamic and static phases separated in time (precision grip). Coherence analysis between local field potentials (LFPs) recorded from neural centers and muscles during the different movement phases revealed: (1) spinal cord and muscles to be coherent across a broad-band of frequencies during both the dynamic and static phases of movement (broad-band coherence); (2) spinal cord and muscles to be coherent at beta-band frequencies (15-30Hz) during the dynamic phase of movement; and (3) motor cortex and muscles to be coherent at beta-band frequencies during the static phase of movement. They also showed differences between these 3 types of coherence in their latencies relative to movement, their muscle distributions, and their

directions of causality. Overall the data support the notion that different spinal and cortical neural networks underlie dynamic and static control of forelimb movements.

Overall impression of the work

These experiments are at the extreme end of difficulty and the Authors should be congratulated for completing and disseminating their work. The research directly addresses several important questions in neural control of movement, it employs sound methods, and it provides concise and novel data relevant to each question with relatively clear outcomes. I support publication in Communications Biology.

Specific comments

1. Parts of the Introduction have sentences that are too long, leading to a lack of clarity.

In particular the consecutive sentences beginning line 33 and ending 42 are both unclear to me. The first sentence I think you mean “Neurons in caudal motor cortex and those with direct connections to spinal motoneurons (e.g. corticomotoneuronal and premotor spinal interneurons) tend to discharge tonically during the static phase.” The point of the second sentence is less clear to me. It seems contradictory since the Authors seem to be saying that the cited studies are deficient in that the effective connectivity of recorded cells to muscles was not assessed, however it was assessed and found to be incongruent with their discharge properties.

Also, the sentence beginning line 45 is long and difficult to follow. Are points 1 and 2 separate questions that should be communicated as such?

We are grateful to the reviewer for appreciating the significance of our study.

According to these comments We have revised this part of the Introduction to clarify the points the reviewer has made (lines 32-46).

2. Table S2. The vast majority of data pertaining to spinal cord LFP and EMG pairs were harvested from 1 of 4 monkeys (monkey E). Despite this I could find no mention anywhere in the manuscript of how well spinal cord LFP and EMG data from the other

3 monkeys matched monkey E. Such a statement is necessary to assess how generalizable the conclusions drawn from the study are across monkeys.

Similarly, although the distribution of data pertaining to motor cortex LFP and EMG pairs is better balanced across the 2 monkeys, a statement about how similar data from each monkey are would be helpful.

We thank for making an important point. We admit that we had failed to describe the results for each monkey. We provide the individual variability in supplemental figure S6. As shown in S6, spinal coherence predominantly during grip phase is common across four monkeys. Likewise, cortical coherence predominantly during hold phase is consistent for two monkeys. We, therefore, pooled the data for the subsequent analyses. We have added these descriptions (in parentheses) to lines 114–127.

3. The Authors rightly point out in the Introduction that neurons in different parts of the motor cortex and spinal cord exhibit discharge patterns better correlated with specific phases of movement (dynamic versus static). The manuscript would be improved if similar analyses were applied to the Authors' data. This would inform upon questions such as: (1) are there any differences between rostral and caudal motor cortex? And (2) are broad-band spinal locations, which are suggested to be more efferently connected with muscles, located more ventrally than narrow-band spinal locations, which are suggested to be more bidirectionally (efferently and afferently) connected?

We thank for insightful comments. Our responses to questions (1) and (2) are described below:

(1) As shown in supplemental figure S4, we recorded mostly in the caudal part, since, from previous reports, corticomuscular coherence is likely to be found in the caudal part. The ratios of grip/hold between rostral and caudal parts are 2/11 (0.18) and 10/80 (0.125), respectively. From the ratio, there is no difference between the sites. If anything, coherence during the grip phase seems to be relatively more frequent in the caudal part. As such, coherence patterns are not likely different between rostral and caudal divisions.

(2) The depth distribution of the spinal BB and NB are now provided in supplementary figure S3. In the figure, while NB coherence was observed throughout the recording sites, BB was found at specific depths relatively ventrally ($p = 0.0034$, chi-squared test).

These depths are likely to correspond to the sites where motoneuron pools of forearm/hand muscles are located in the cervical enlargement. We have added these observations in lines 206–210.

Also, the cortical recording chambers were located over the pre- and post-central gyri. Is there any data from LFP recordings in post-central gyrus? This may help support and develop the conclusions drawn.

We thank for providing a very insightful point. It is very interesting to examine the coherence between the somatosensory cortex and EMGs for comparison with that of the motor cortex. However, we focused on motor-related structures in this study, and hence did not record neural activity from the somatosensory cortex. This is definitely warranted for further studies.

References

- [1] T. D. Aumann and Y. Prut. Do sensorimotor beta-oscillations maintain muscle synergy representations in primary motor cortex? *Trends in Neurosciences*, 38(2):77–85, 2015.
- [2] S. N. Baker, J. M. Kilner, E. M. Pinches, and R. N. Lemon. The role of synchrony and oscillations in the motor output. *Experimental Brain Research*, 128(1-2):109–117, 1999.
- [3] S. N. Baker, E. Olivier, and R. N. Lemon. Coherent oscillations in monkey motor cortex and hand muscle EMG show task-dependent modulation. *The Journal of Physiology*, 501(1):225–241, 1997.
- [4] S. N. Baker, R. Spinks, A. Jackson, and R. N. Lemon. Synchronization in monkey motor cortex during a precision grip task. I. Task-dependent modulation in single-unit synchrony. *Journal of Neurophysiology*, 85(2):869–885, 2001.
- [5] J. Confais, G. Kim, S. Tomatsu, T. Takei, and K. Seki. Nerve-Specific Input Modulation to Spinal Neurons during a Motor Task in the Monkey. *Journal of Neuroscience*, 37(10):2612–2626, 2017.

- [6] M. Denker, S. Roux, H. Lind en, M. Diesmann, A. Riehle, and S. Grun. The Local Field Potential Reflects Surplus Spike Synchrony. *Cerebral Cortex*, 21(12):2681–2695, 2011.
- [7] E. E. Fetz and P. D. Cheney. Postspike facilitation of forelimb muscle activity by primate corticomotoneuronal cells. *Journal of Neurophysiology*, 44(4):751–772, 1980.
- [8] R. Jurgens, W. Becker, and H. H. Kornhuber. Natural and drug-induced variations of velocity and duration of human saccadic eye movements: evidence for a control of the neural pulse generator by local feedback. *Biological Cybernetics*, 39(2):87–96, 1981.
- [9] I. Kurtzer, T. M. Herter, and S. H. Scott. Random change in cortical load representation suggests distinct control of posture and movement. *Nature Neuroscience*, 8(4):498–504, 2005.
- [10] M. A. Maier, K. M. Bennett, M. C. Hepp-Reymond, and R. N. Lemon. Contribution of the monkey corticomotoneuronal system to the control of force in precision grip. *Journal of Neurophysiology*, 69(3):772–785, 1993.
- [11] V. N. Murthy and E. E. Fetz. Synchronization of neurons during local field potential oscillations in sensorimotor cortex of awake monkeys. *Journal of Neurophysiology*, 76(6):3968–3982, 1996.
- [12] Y. Prut and S. I. Perlmutter. Firing properties of spinal interneurons during voluntary movement. II. Interactions between spinal neurons. *The Journal of Neuroscience*, 23(29):9611–9619, 2003.
- [13] A. Riehle, S. Grun, M. Diesmann, and A. Aertsen. Spike synchronization and rate modulation differentially involved in motor cortical function. *Science*, 278(5345):1950–1953, 1997.
- [14] D. A. Robinson. Models of the saccadic eye movement control system. *Kybernetik*, 14(2):71–83, 1973.

- [15] K. Seki and E. E. Fetz. Gating of Sensory Input at Spinal and Cortical Levels during Preparation and Execution of Voluntary Movement. *Journal of Neuroscience*, 32(3):890–902, 2012.
- [16] R. Shadmehr. Distinct neural circuits for control of movement vs. holding still. *Journal of Neurophysiology*, 117(4):1431–1460, 2017.
- [17] S. W. Smith. *The Scientist and Engineer’s Guide to Digital Signal Processing*. California Technical Pub, 1999.
- [18] T. Takei and K. Seki. Spinomuscular Coherence in Monkeys Performing a Precision Grip Task. *Journal of Neurophysiology*, 99(4):2012–2020, 2008.
- [19] T. Takei and K. Seki. Spinal Interneurons Facilitate Coactivation of Hand Muscles during a Precision Grip Task in Monkeys. *Journal of Neuroscience*, 30(50):17041–17050, 2010.
- [20] T. Takei and K. Seki. Spinal Premotor Interneurons Mediate Dynamic and Static Motor Commands for Precision Grip in Monkeys. *Journal of Neuroscience*, 33(20):8850–8860, 2013.
- [21] T. Womelsdorf, J. M. Schoffelen, R. Oostenveld, W. Singer, R. Desimone, A. K. Engel, and P. Fries. Modulation of Neuronal Interactions Through Neuronal Synchronization. *Science*, 316(5831):1609–1612, 2007.

REVIEWERS' COMMENTS:

Reviewer #1 (Remarks to the Author):

The authors have responded nicely to all of my major concerns. I have added a few comments here that I think will improve clarity.

24: Similar control processes for each action has / specialized control processes have
"Control processes" requires a plural verb (have). I would suggest instead to state, "Use of a single control process for both moving and holding has..."

32 It remain unknown
remains

49 during the sustained control of muscle action but not during the dynamic phase
during the sustained control of muscle action, not during the dynamic phase

275 supposedly act cooperatively to achieve an efficient neuronal transmission.
"supposedly" carries the connotation that the statement is suspect. I'd state instead, "are thought".

276 is proposed that enhanced efficacy through synchronized oscillation may contribute to reducing
"may contribute to reduced"

280 mechanism supposedly might act
Delete "supposedly"

282 the hold period may be related to this decorrelated mechanism
to this decorrelation mechanism

1D: I would suggest again that you consider a different title for these plots that does not refer to
"coherence", which is only relevant for the lowest set of panels. Also, the font size too small
(particularly for the exponents).

486 S2). The latency for each pair was calculated as medians of the first time that rises above
significant levels
We computed the latency for each pair based on the median of the distribution of times for which
broadband coherence exceeded a given significance level.

Fig 3: Why so little radial nerve coherence? How well does EMG magnitude predict coherence?
The authors are correct that EMG normalization using MVC in a monkey is not terribly reliable, but a
comparison of this effect across muscles is not necessary. Looking for a relation within muscles would
be equally (or more) informative. S5 does nicely illustrate that the extensors were active in this task,
even (at least for EDC) during the hold period, making this comparison less important. Some
reference to this radial nerve puzzle would be warranted.

189 involved in each coherence are also distinct; the trans-spinal loop involves the index finger
muscles (FDI and FDP_r) with interactions among local forearm extensors, whereas the trans-cortical
feedback loop arise largely through the recruited finger muscles,
I would suggest a further modification of this sentence, for greater clarity: "...the trans-spinal loop
involves interactions between the index finger muscles (FDI and FDP_r) and neighboring forearm
extensors, whereas the trans-cortical feedback loop arises more broadly through all the recruited

finger muscles,

Reviewer #2 (Remarks to the Author):

The authors have extensively revised the manuscript and responded to all my previous comments comprehensively and convincingly. I have no further comments.

Reviewer #3 (Remarks to the Author):

The Authors have adequately addressed all of my concerns.

We have attached a pdf file ("main_markup.pdf") in which line numbers are provided and revised texts are colored in red.

Reviewer #1 (Remarks to the Author):

The authors have responded nicely to all of my major concerns. I have added a few comments here that I think will improve clarity.

24: Similar control processes for each action has / specialized control processes have
"Control processes" requires a plural verb (have). I would suggest instead to state, "Use of a single control process for both moving and holding has..."

We have corrected the phrase accordingly in lines 4-6.

32 It remain unknown
remains

We have corrected the phrase accordingly in line 10.

49 during the sustained control of muscle action but not during the dynamic phase
during the sustained control of muscle action, not during the dynamic phase

We have corrected the phrase accordingly in lines 21-22.

275 supposedly act cooperatively to achieve an efficient neuronal transmission.
"supposedly" carries the connotation that the statement is suspect. I'd state instead, "are thought".

We have corrected the phrase accordingly in line 200.

276 is proposed that enhanced efficacy through synchronized oscillation may contribute to reducing
"may contribute to reduced"

We have corrected the phrase accordingly in line 201.

280 mechanism supposedly might act
Delete "supposedly"

We have corrected the phrase accordingly in line 203.

282 the hold period may be related to this decorrelated mechanism
to this decorrelation mechanism

We have corrected the phrase accordingly in lines 205-206.

1D: I would suggest again that you consider a different title for these plots that does not refer to "coherence", which is only relevant for the lowest set of panels. Also, the font size too small (particularly for the exponents).
We have corrected the legend as "c-e Representative patterns of power spectra of neural LFPs, EMGs and their coherence between spinal broad-band (BB) LFP and AbPB (c) spinal narrow-band (NB) LFP and AbPL (d), and cortical NB LFP and AbDM (e)". We have amended the font size.

486 S2). The latency for each pair was calculated as medians of the first time that rises above significant levels
We computed the latency for each pair based on the median of the distribution of times for which broadband coherence exceeded a given significance level.

We have corrected the phrase accordingly in lines 368-370.

Fig 3: Why so little radial nerve coherence? How well does EMG magnitude predict coherence?

The authors are correct that EMG normalization using MVC in a monkey is not terribly reliable, but a comparison of this effect across muscles is not necessary. Looking for a relation within muscles would be equally (or more) informative. S5 does nicely illustrate that the extensors were active in this task, even (at least for EDC) during the hold period, making this comparison less important. Some reference to this radial nerve puzzle would be warranted.

We have added the description on the radial-innervated muscles in lines 87-88, and reference to the puzzle in discussion in lines 234-237.

189 involved in each coherence are also distinct; the trans-spinal loop involves the index finger muscles (FDI and FDP_r) with interactions among local forearm extensors, whereas the trans-cortical feedback loop arise largely through the recruited finger muscles,

I would suggest a further modification of this sentence, for greater clarity: "...the trans-spinal loop involves interactions between the index finger muscles (FDI and FDP_r) and neighboring forearm extensors, whereas the trans-cortical feedback loop arises more broadly through all the recruited finger muscles,

We have corrected the phrase accordingly in lines 136-139.